Corrected: Author correction

# Blunting neuroinflammation with resolvin D1 prevents early pathology in a rat model of Parkinson's disease

Paraskevi Krashia [1,2,11], Alberto Cordella[1,3,11], Annalisa Nobili[1,2,11], Livia La Barbera[1,3], Mauro Federici [1], Alessandro Leuti[1,2], Federica Campanelli[1], Giuseppina Natale[1], Gioia Marino[1], Valeria Calabrese [1], Francescangelo Vedele[1,3], Veronica Ghiglieri[1,4], Barbara Picconi[1], Giulia Di Lazzaro[3], Tommaso Schirinzi[3], Giulia Sancesario[5], Nicolas Casadei[6], Olaf Riess[6], Sergio Bernardini[7], Antonio Pisani [1,3], Paolo Calabresi[1,8], Maria Teresa Viscomi[9], Charles Nicholas Serhan [10], Valerio Chiurchiù[1,2,12], Marcello D'Amelio[1,2,12] & Nicola Biagio Mercuri [1,3,12]

Neuroinflammation is one of the hallmarks of Parkinson's disease (PD) and may contribute to midbrain dopamine (DA) neuron degeneration. Recent studies link chronic inflammation with failure to resolve early inflammation, a process operated by specialized pro-resolving mediators, including resolvins. However, the effects of stimulating the resolution of inflammation in PD – to modulate disease progression – still remain unexplored. Here we show that rats overexpressing human α-synuclein (Syn) display altered DA neuron properties, reduced striatal DA outflow and motor deficits prior to nigral degeneration. These early alterations are coupled with microglia activation and perturbations of inflammatory and pro-resolving mediators, namely IFN-γ and resolvin D1 (RvD1). Chronic and early RvD1 administration in Syn rats prevents central and peripheral inflammation, as well as neuronal dysfunction and motor deficits. We also show that endogenous RvD1 is decreased in human patients with early-PD. Our results suggest there is an imbalance between neuroinflammatory and pro-resolving processes in PD.

[1] Department of Experimental Neurosciences, IRCCS Santa Lucia Foundation, 00143 Rome, Italy. [2] Department of Medicine and Department of Science and Technology for Humans and Environment, University Campus Bio-medico, 00128 Rome, Italy. [3] Department of Systems Medicine, University of Rome 'Tor Vergata', 00133 Rome, Italy. [4] Department of Philosophy, Human, Social and Educational Sciences, University of Perugia, 06123 Perugia, Italy. [5] Department of Clinical and Behavioural Neurology, IRCCS Santa Lucia Foundation, 00143 Rome, Italy. [6] Institute of Medical Genetics and Applied Genomics, University of Tübingen, 72076 Tübingen, Germany. [7] Department of Experimental Medicine and Surgery, University of Rome 'Tor Vergata', 00133 Rome, Italy. [8] Neurology Clinic, Department of Medicine, University of Perugia, Santa Maria della Misericordia Hospital, 06156 Perugia, Italy. [9] Institute of Histology and Embryology, Università Cattolica del Sacro Cuore, 00168, Rome, Italy. [10] Center for Experimental Therapeutics and Reperfusion Injury, Department of Anaesthesiology, Perioperative and Pain Medicine, Brigham and Women's Hospital and Harvard Medical School, 02115 Boston, MA, USA. [11]These authors contributed equally: Paraskevi Krashia, Alberto Cordella, Annalisa Nobili. [12]These authors jointly supervised this work: Valerio Chiurchiù, Marcello D'Amelio, Nicola Biagio Mercuri. Correspondence and requests for materials should be addressed to N.B.M. (email: mercurin@med.uniroma2.it)

Parkinson's disease (PD) is a neurodegenerative disorder characterized by motor and non-motor symptoms including tremor, rigidity, bradykinesia, postural instability, constipation and depression. Although several neuronal populations are affected in PD, the principal underlying pathophysiology is determined by degeneration of substantia nigra pars compacta (SNpc) dopamine (DA) neurons, leading to impaired dopaminergic neurotransmission in the dorsolateral striatum[1].

Most PD cases are sporadic, featured by neurites expressing α-synuclein (α-syn)-rich Lewy bodies[1,2]. There is overwhelming evidence that α-syn mutations are determinant of both familial and sporadic PD[2–4], while overexpression of non-mutated α-syn can also increase the risk for PD[5,6]. However, our understanding of how increased α-syn levels can drive the events leading to PD is incomplete. Today, there is general agreement that many other factors, either alone or in combinations, can contribute to the neurodegenerative processes, including DA metabolism, mitochondrial dysfunction, oxidative stress, impaired protein degradation and neuroinflammation[7–12].

Indeed, neuroinflammation is a well-established feature of PD[12–15] and aggregated α-syn can induce microglia activation and inflammatory cytokine release much earlier than the occurrence of DA cell death[16,17]. In fact, the SNpc not only shows high density of microglia[18,19] but evidence from experimental models suggest that DA neurons are highly and selectively vulnerable to inflammatory attacks[19–22], supporting the hypothesis that the microglia-mediated neuroinflammation contributes to the cascade of events that lead to degeneration and worsening of the disease[12,23]. Neuroinflammation could be a consequence of failure to resolve inflammation and to restore tissue homeostasis, the resolution of which is mediated by specialized pro-resolving lipid mediators, a superfamily of pro-resolving lipids that derive metabolically from ω-3 and ω-6 essential fatty acids[24–26]. However, the effect of modulating such neuroinflammatory circuits, in order to reverse or slow down the disease progression, is yet unexplored.

To address this question and shed light into the relationship between α-syn load and inflammation, we used a validated transgenic rat model of PD that overexpresses the human non-mutated α-syn (Syn rats)[27]. Here we show that α-syn overexpression in Syn rats induces early alterations (i.e. long before DA neuron degeneration) in DA neuron properties, nigrostriatal dopaminergic transmission and motor behaviour. We also provide evidence for a strong link between α-syn overexpression and neuroinflammation, showing a reduction of resolvin D1 (RvD1), a specific pro-resolving mediator. Remarkably, early chronic treatment of Syn rats with RvD1 reduces neuroinflammation, restores dopaminergic neurotransmission and prevents development of neuronal deficits and motor impairment. We also report a central and peripheral RvD1 impairment in early-PD patients. Taken together, our findings reveal that boosting the resolution of inflammation can prevent early α-syn-induced neuroinflammation, neurophysiological and motor deficits, suggesting that resolvins could be therapeutically exploited as novel diagnostic biomarkers and disease-modifying agents.

## Results

**Early reduction of striatal DA in Syn rats.** Given that α-syn overexpression is pathogenic in PD and other synucleinopathies, we first characterized the very early effects of α-syn overexpression on the midbrain DA system, to highlight factors that contribute to DA neuron dysfunction and ultimately to cell death. We used a recently developed BAC transgenic rat model (Syn rats)[27] that overexpresses the full-length human α-syn and recapitulates common PD features such as widespread α-syn aggregation, progressive DA cell loss, loss of projecting fibres in the striatum and associated motor symptoms[27,28].

To identify the earliest timepoint of motor impairment in Syn rats, we analysed two different age groups: asymptomatic 2-month-old animals and 4-month-old rats with early anxiety-like behaviour. In agreement with earlier reports[27,28], 4-month-old Syn rats showed normal basal locomotor activity in the open field compared to aged-matched wild-type (WT) animals but performed fewer entries to the centre zone, indicating increased anxiety-like behaviour as avoidance of the centre (Fig. 1a). This behaviour was age-dependent since 2-month-old Syn animals behaved similarly to controls. We then subjected rats to an accelerating rotarod test, a more complex test requiring motor coordination and learning, to highlight the subtle motor deficits observed in young Syn rats[27,28]. Four-month-old Syn rats showed significant motor impairment compared to age-matched WT, which was absent in younger Syn animals (Fig. 1b). These data are in line with an age-dependent symptom progression and demonstrate that the 2–4-month age-window is ideal for studying early effects of α-syn overexpression on the nigrostriatal system.

In line with the motor impairment, 4-month-old Syn rats showed significant reduction of DA outflow in the dorsal striatum, while 2-month-old animals showed normal DA levels (Fig. 1c). The reduced striatal DA was not associated with changes in the integrity of DA-releasing terminals, measured with tyrosine hydroxylase (TH) labelling (Fig. 1d), nor was it due to DA release deficits in the presence of amphetamine (Fig. 1e). Additionally, stereological cell counts in the SNpc and the neighbouring ventral tegmental area (VTA) showed similar TH$^+$ neuron numbers between Syn and WT animals (Fig. 1f).

Altogether, these data show that human α-syn overexpression causes early functional deficits in nigrostriatal neurotransmission. As in other PD models[29], in Syn rats these deficits precede DA neuron degeneration or striatal denervation and are temporally associated with motor symptoms.

**Altered properties of SNpc DA neurons in Syn rats.** Given that the reduced DA outflow in the striatum was not due to degeneration or denervation, we looked for changes in the functional properties of SNpc DA neurons in Syn rats by performing a detailed electrophysiological characterization in 2- and 4-month-old animals.

In 4-month-old Syn rats the DA neuron firing frequency was reduced, accompanied by reduction in the regularity of firing, measured as increase in the coefficient of variation of inter-spike interval (CV-ISI; Fig. 2a). On the contrary, 2-month-old Syn rats showed normal pacemaker activity (Supplementary Fig. 1a). Similarly, the number of action potential (APs) in Syn neurons, generated by depolarizing current steps during current-clamp, was reduced compared to WT neurons (Fig. 2b), while DA cells in 2-month-old Syn animals fired APs similarly to WT (Supplementary Fig. 1b). The reduced excitability of 4-month-old Syn neurons was not due to differences in membrane resistance (Rm) or sag (Fig. 2c and Supplementary Fig. 2a), nor due to changes in cell capacitance (Cm; Fig. 2d) and threshold potential (Supplementary Fig. 2b). Thus, we next sought to examine characteristic DA neuron conductances, such as the hyperpolarization-activated current ($I_h$)[30,31]. In agreement with the fact that the sag was unchanged in Syn neurons, the $I_h$ current amplitude and activation kinetics were also unchanged (Fig. 2e). However, the after-hyperpolarization potential reached during pacemaking was more negative in Syn rats (Supplementary Fig. 2b, c), suggesting an increase in the after-hyperpolarization current (AHC). Indeed, depolarization to 0 mV induced a stronger, faster-deactivating AHC in Syn rats (Fig. 2f), largely

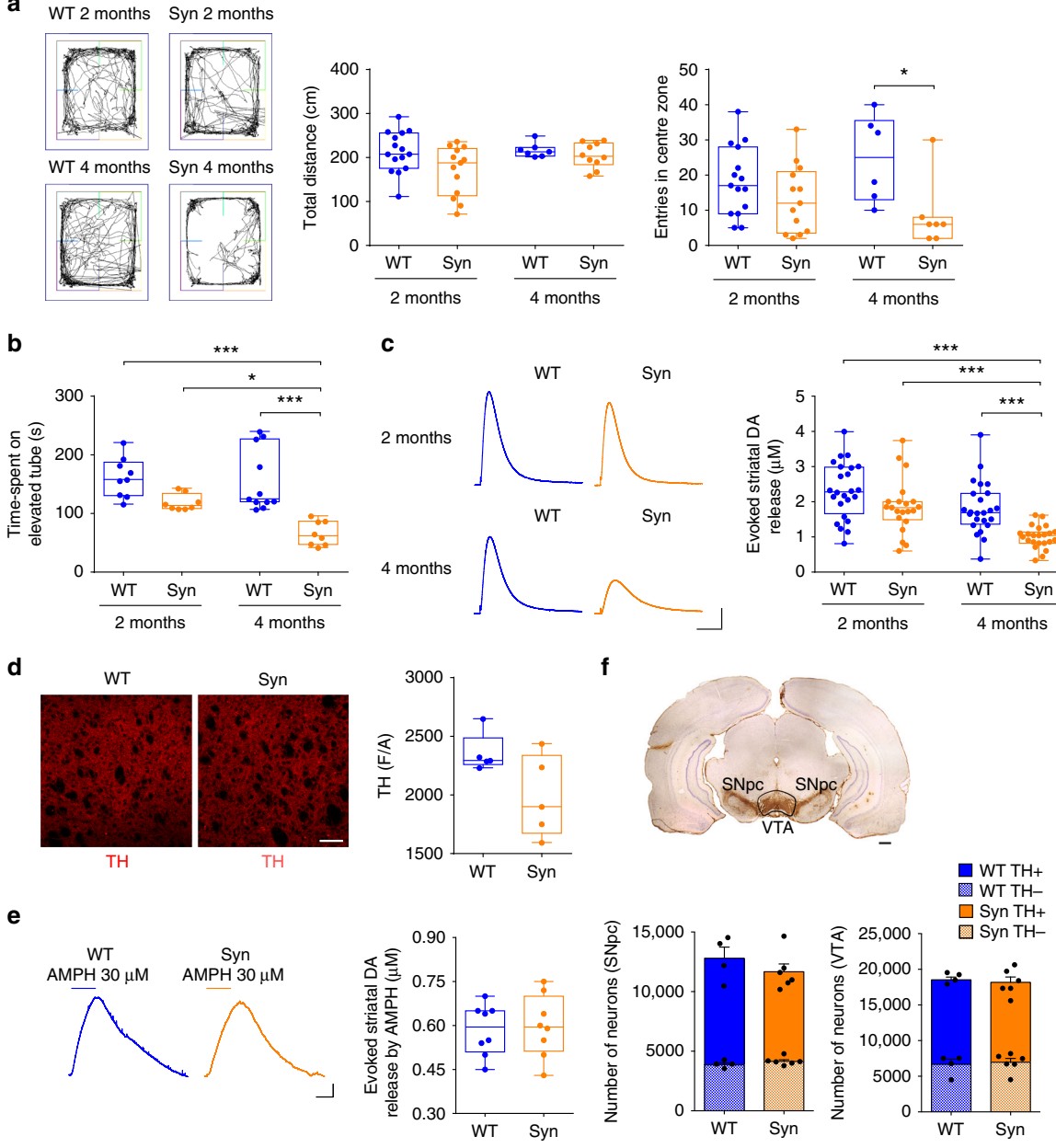

**Fig. 1** Motor deficits and reduced striatal DA in 4-month-old Syn rats. **a** Distance travelled during open field test (2-month-old: 15 WT, 13 Syn; 4-month-old: 7 WT, 10 Syn; two-way ANOVA: genotype × age, $F_{1,41} = 0.10$, $P = 0.748$; genotype, $F_{1,41} = 6.77$, $P = 0.013$; age, $F_{1,41} = 4.70$, $P = 0.037$; $P > 0.05$ with Bonferroni's) and centre zone entries (2-month-old: 15 WT, 13 Syn; 4-month-old: 6 WT, 7 Syn; two-way ANOVA: genotype × age, $F_{1,37} = 2.65$, $P = 0.112$; genotype, $F_{1,37} = 9.66$, $P = 0.004$; age, $F_{1,37} = 0.06$, $P = 0.799$; WT 4-Syn 4 *$P = 0.041$ with Bonferroni's). **b** Time spent on the accelerating Rotarod (2-month-old: 9 WT, 8 Syn; 4-month-old: 11 WT, 8 Syn; two-way ANOVA: genotype × age, $F_{1,32} = 3.79$, $P = 0.061$; genotype, $F_{1,32} = 29.69$, $P < 1.00 \times 10^{-4}$; age, $F_{1,32} = 5.91$, $P = 0.021$; WT 2 months-Syn 4 months ***$P < 1.00 \times 10^{-4}$, Syn 2 months-Syn 4 months *$P = 0.029$, WT 4 months-Syn 4 months ***$P < 1.00 \times 10^{-4}$ with Bonferroni's). **c** Example traces (scale: 50 pA, 500 ms) and evoked DA in the striatum (2-month-old: 24 WT, 21 Syn slices, 4 rats each; 4-month-old: 24 WT, 23 Syn slices, 4 rats each; two-way ANOVA: genotype × age, $F_{1,88} = 1.72$, $P = 0.193$; genotype, $F_{1,88} = 19.95$, $P < 1.00 \times 10^{-4}$; age, $F_{1,88} = 22.50$, $P < 1.00 \times 10^{-4}$; WT 2-Syn 4 ***$P < 1.00 \times 10^{-4}$, Syn 2-Syn 4 ***$P = 4.00 \times 10^{-4}$, WT 4-Syn 4 ***$P = 5.00 \times 10^{-4}$ with Bonferroni's). **d** TH labelling in the striatum (scale: 200 μm) and TH levels (5 rats each, 4 sections per animal; Mann–Whitney, $P = 0.151$). **e** Striatal DA release by amphetamine (AMPH, 30 μM; scale: 10 pA, 5 min) in 4-month-old rats (8 slices, 4 rats per genotype; Welch's $t$-test, $P = 0.802$). **f** TH immunoreactivity in a WT rat (scale: 500 μm) and mean TH$^+$ and TH$^-$ cell numbers (±s.e.m.) in 4-month-old WT (4) and Syn (6) rats (SNpc: two-way ANOVA: genotype × cell number, $F_{1,16} = 2.38$, $P = 0.143$; genotype, $F_{1,16} = 1.02$, $P = 0.327$; numbers, $F_{1,16} = 57.06$, $P < 1.00 \times 10^{-4}$. WT-Syn for TH$^-$: $P > 0.05$; for TH$^+$: $P > 0.05$ with Bonferroni's; VTA: two-way ANOVA: genotype × cell numbers, $F_{1,16} = 0.52$, $P = 0.483$; genotype, $F_{1,16} = 0.07$, $P = 0.792$; numbers, $F_{1,16} = 51.19$, $P < 1.00 \times 10^{-4}$. WT-Syn for TH$^-$: $P > 0.05$; for TH$^+$: $P > 0.05$ with Bonferroni's). In this and all other figures, in box-and-whisker plots the centre lines denote median values, edges are upper and lower quartiles, whiskers show minimum and maximum values and points are individual experiments. Source data are provided as a Source Data file

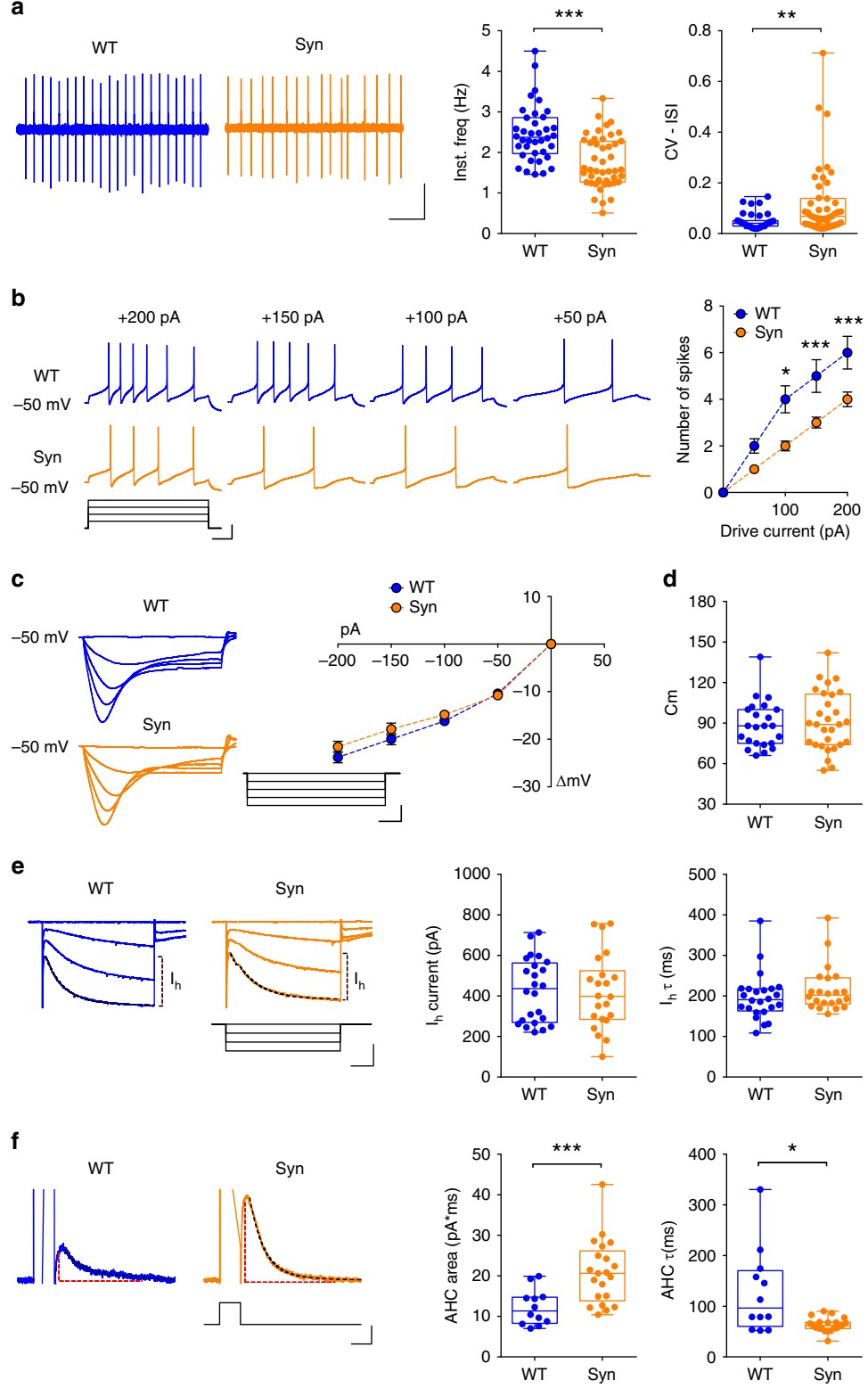

mediated by apamin-sensitive small-conductance $Ca^{2+}$-activated $K^+$ (SK) channels (Supplementary Fig. 2d). These differences between Syn and WT rats were absent from 2-month-old animals (Supplementary Fig. 1c, d).

We next investigated the autoreceptor-mediated inhibition of DA neurons. DA, via somatodendritic D2 autoreceptors, inhibits midbrain DA neurons by activating G-protein-coupled inwardly-rectifying $K^+$ channels (GIRK/Kir3)[32,33]. DA application (2 min,

30 μM) during extracellular recordings caused a lower inhibition of firing in 4-month-old Syn rat neurons compared to WT (Fig. 3a), while no differences were observed between 2-month-old rats across genotypes (Supplementary Fig. 1e). Similarly, DA induced smaller outward currents in Syn neurons during voltage–clamp (−60 mV; Fig. 3b). To decipher whether the reduced sensitivity to DA involved D2 receptors or the downstream GIRK/Kir3 channels, we repeated dose-response experiments using baclofen,

**Fig. 2** Altered DA neuron electrophysiological properties in 4-month-old Syn rats. **a** Spontaneous firing in SNpc DA neurons from 4-month-old rats (scale: 2 min, 0.2 mV) and plots showing firing frequency (38 cells from 10 WT, 47 cells from 11 Syn; Mann–Whitney test, ***$P < 1.00 \times 10^{-4}$) and coefficient of variation of interspike interval (35 cells from 10 WT, 45 cells from 11 Syn; Mann–Whitney test, **$P = 0.002$). **b** APs induced by depolarization (scale: 100 ms; 20 mV, 100 pA) from DA neurons held at −50 mV and mean (±s.e.m.) AP numbers (11 WT, 10 Syn neurons, 3 rats each; two-way repeated-measures ANOVA: genotype×current, $F_{4,76} = 9.36$, $P < 1.00 \times 10^{-4}$; current, $F_{4,76} = 116.6$, $P < 1.00 \times 10^{-4}$; genotype, $F_{1,19} = 10.76$, $P = 0.004$. WT vs Syn: 200 pA ***$P < 1.00 \times 10^{-4}$, 150 pA ***$P = 4.00 \times 10^{-4}$, 100 pA *$P = 0.036$, 50 pA $P = 0.221$ with Bonferroni's). **c** Sub-threshold responses to hyperpolarization (scale: 100 ms; 10 mV, 100 pA) and current/voltage plots (±s.e.m.; 11 WT, 10 Syn neurons, 3 rats each; two-way repeated-measures ANOVA: genotype×current, $F_{4,76} = 1.64$, $P = 0.172$; current, $F_{4,76} = 362.9$, $P < 1.00 \times 10^{-4}$; genotype, $F_{1,19} = 1.26$, $P = 0.276$. WT vs Syn: 0 pA $P > 0.999$, −50 pA, $P > 0.999$, −100 pA, $P > 0.999$, −150 pA, $P = 0.496$, −200 pA, $P = 0.411$ with Bonferroni's). **d** Cm (23 WT, 29 Syn neurons, 5 rats each: Welch's $t$-test $P = 0.638$). **e** $I_h$ currents from DA neurons held at −60 mV and hyperpolarized in 20 mV increments (scale: 200 ms; 50 mV, 200 pA). The activation phase was fitted to 1–2 exponentials (dash line). The plots show $I_h$ amplitude and weighted activation tau after hyperpolarization to −120 mV (24 WT, 22 Syn neurons, 8 rats each; amplitude: Welch's $t$-test $P = 0.886$; tau: Mann–Whitney test $P = 0.342$). **f** AHC (scale: 100 ms; 50 mV, 50 pA), induced by depolarization to 0 mV. AHC trace area is designated by red dash lines and deactivation is obtained with exponential fits (12 WT, 21 Syn neurons, 4 rats each; area: Welch's $t$-test ***$P = 3.00 \times 10^{-4}$; tau: Mann–Whitney test *$P = 0.013$). Source data are provided as a Source Data file

an agonist of GABA$_B$ receptors that, similarly to D2 receptors, are expressed in SNpc neurons and mediate their action via GIRK/Kir3 channels[33–35]. Baclofen currents were smaller in 4-month-old Syn rats (Fig. 3c), confirming that GIRK/Kir3 channel function in DA neurons is impaired.

Overall, the alterations in K$^+$ conductances, apamin-sensitive AHC and cell firing in Syn DA neurons, at an age preceding cell degeneration, are all indications of cellular dysfunction. Accordingly, microfluorometry measurements of somatic cytoplasmic [Ca$^{2+}$] at −60 mV (to prevent firing) resulted in significantly higher [Ca$^{2+}$] in DA neurons from 4-month-old Syn rats (Fig. 3d), whereas no changes were detected in 2-month-old animals (Supplementary Fig. 1f). Since SK channel opening is particularly sensitive to cytoplasmic [Ca$^{2+}$] changes[36], the abnormal Ca$^{2+}$ accumulation in Syn DA neurons can explain the observed rise in the apamin-sensitive AHC (Fig. 2f)[37–40]. To investigate the source of high Ca$^{2+}$ we performed microfluorometry measurements in the presence of Ca$^{2+}$-free extracellular solution, to analyse the contribution of voltage-gated Ca$^{2+}$ channels (VGCCs), or in the presence of isradipine, an antagonist of L-type VGCCs. Ca$^{2+}$ entry via somatodendritic L-type VGCCs during pacemaking increases mitochondrial oxidative stress in vulnerable DA neurons and has been strongly implicated in PD[11,41]. The application of Ca$^{2+}$-free extracellular solution reduced cytoplasmic [Ca$^{2+}$] in both WT and Syn neurons, but at a comparable extent, and the difference in Ca$^{2+}$ levels between genotypes was still evident following 4 min in Ca$^{2+}$-free conditions (Fig. 3e). Similar results were obtained after incubating midbrain slices with isradipine (1 h, 200 nM), which failed to cancel the difference between WT and Syn neurons (Fig. 3f), suggesting that at −60 mV the contribution of VGCCs to the higher [Ca$^{2+}$] in Syn DA neurons is minimal. To investigate the contribution of endoplasmic reticulum (ER) Ca$^{2+}$ stores instead, we used cyclopiazonic acid (CPA), a potent blocker of the SERCA Ca$^{2+}$ pump that refills ER Ca$^{2+}$ stores. CPA causes a fast depletion of ER stores due to Ca$^{2+}$ leakage to the cytoplasm, evident as a transient increase in cytoplasmic [Ca$^{2+}$] (Fig. 3g), providing an estimate of the Ca$^{2+}$ accumulated in the stores. In Syn DA neurons bath application of 10 μM CPA resulted in a lower Ca$^{2+}$ transient compared to WT neurons (Fig. 3g), suggesting lower Ca$^{2+}$ content in CPA-sensitive stores and impaired storing capacity that could explain the cytosolic Ca$^{2+}$ accumulation in Syn DA neurons.

These data confirm that α-syn overexpression leads to multiple functional alterations in SNpc DA neurons that can overall contribute to the observed age-dependent degeneration in older animals[27].

**Central and peripheral inflammation in Syn rats.** Given the link between neuroinflammation and α-syn pathology in PD[16,17,42,43],

we next investigated signs of neuroinflammation in 4-month-old Syn rats. We focused on four different brain areas, the SNpc, dorsolateral striatum, dorsal hippocampus and pontine nuclei, to examine whether α-syn overexpression was associated with either increased numbers or reactivity of astrocytes and microglia. Analysis of glial fibrillary acidic protein-positive (GFAP$^+$) astrocytes in Syn rats did not show differences in either cell number or morphology compared to WT rats in any of the brain areas (Fig. 4a, c, d and Supplementary Fig. 3). Instead, we observed an increase in ionized calcium binding adaptor protein-positive (Iba1$^+$) microglia numbers in the SNpc, striatum (Fig. 4b) and dorsal hippocampus of Syn rats, but microglia numbers were unchanged in the pontine nuclei (Supplementary Fig. 4). The increase in Iba1$^+$ cell numbers was also accompanied by morphological changes, whereby microglia displayed a higher degree of complexity, increased number of intersections (Fig. 4e, f), thicker and more branched processes, longer ramifications and increased number of nodes compared to WT rats (Supplementary Fig. 4).

We next analysed the levels of several pro-inflammatory (tumour necrosis factor, TNF-α; interferon-γ, IFN-γ; interleukins IL-1β and IL-6) and anti-inflammatory (IL-10, IL-4 and IL-13) cytokines in the cerebrospinal fluid (CSF) and plasma. Although no detectable cytokines were found in the plasma of either 2- or 4-month-old animals, we found higher levels of IFN-γ in the CSF of Syn rats (Fig. 4g). This increase was age-dependent, being higher in 4- compared to 2-month-old Syn rats. Additionally, we observed a trend for lower CSF levels of the anti-inflammatory IL-10 in Syn rats ($3.02 \pm 1.10$ pg ml$^{-1}$ in WT vs $0.50 \pm 0.50$ pg ml$^{-1}$ in 2-month-old Syn rats, $P = 0.143$ with Mann–Whitney test; $4.48 \pm 0.52$ pg ml$^{-1}$ in WT vs $2.43 \pm 0.57$ pg ml$^{-1}$ in 4-month-old Syn rats, $P = 0.151$ with Welch's $t$-test).

Since α-syn and microglia via IFN-γ can interact to trigger DA neuron loss in PD[44–46], we tested the acute IFN-γ effect on striatal DA release in 2-month-old slices, to investigate whether the combined presence of the two pro-inflammatory triggers (α-syn overexpression and IFN-γ) would accelerate the appearance of deficits in 2-month-old animals that otherwise show normal DA release (Fig. 1c). Incubation of striatal slices with IFN-γ (100–200 ng ml$^{-1}$, for 3 h) caused a strong reduction in striatal DA release in 2-month-old Syn rats that was comparable to that seen in 4-month-old animals (Supplementary Fig. 5a), suggesting that the cytokine can accelerate the appearance of DA release deficits in Syn rats. IFN-γ had no effect on WT slices.

To investigate whether also peripheral immune events are involved, we analysed the numbers of leukocytes in 4-month-old rats using polychromatic flow cytometry (for gating strategy see Supplementary Fig. 6a). While the numbers of granulocytes, T and B lymphocytes remained unchanged between genotypes, monocytes were significantly reduced in Syn rats (Fig. 4h) and showed

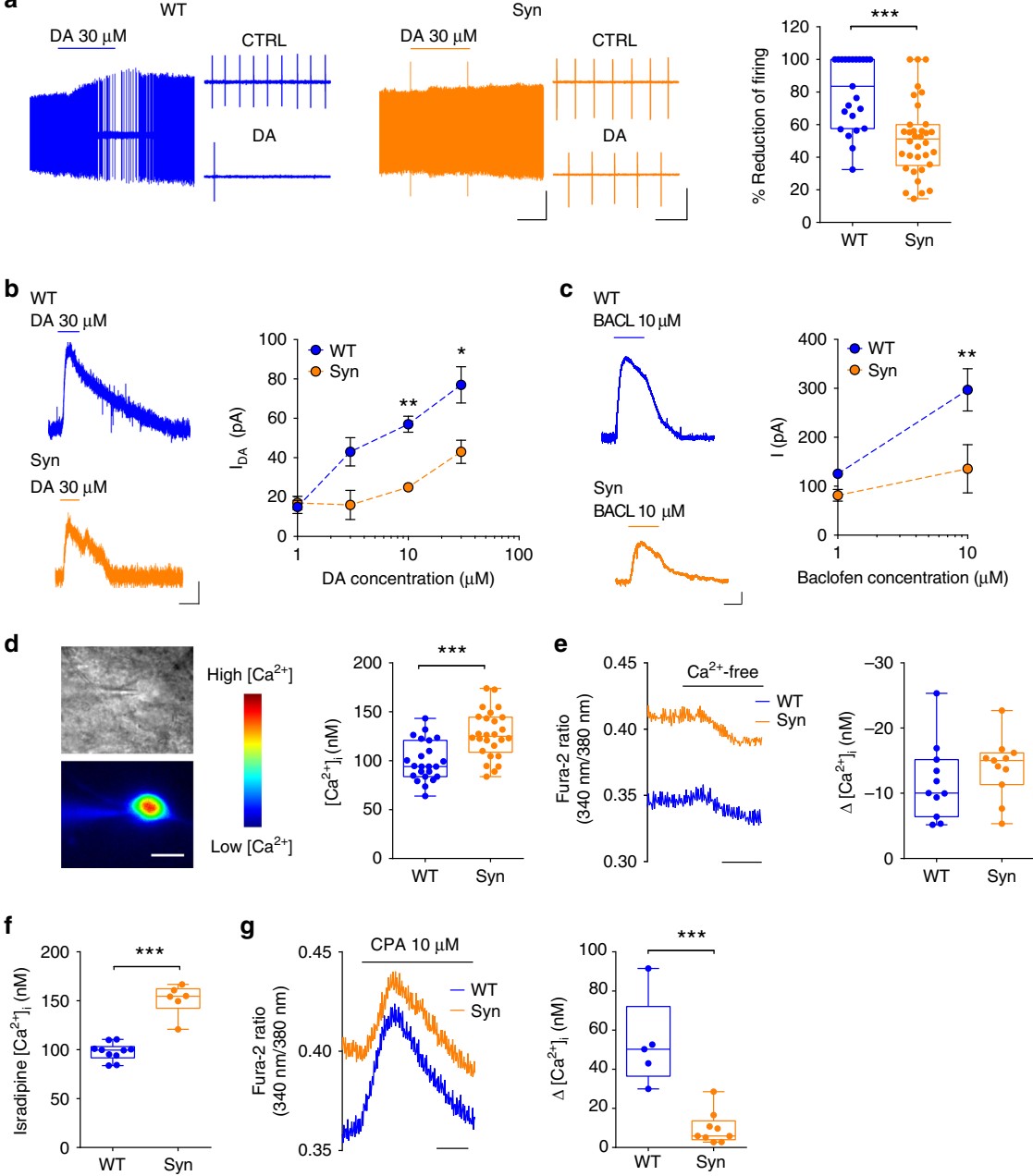

**Fig. 3** DA sensitivity and intracellular [Ca$^{2+}$] are altered in 4-month-old Syn DA neurons. **a** Extracellular recordings in SNpc DA neurons from 4-month-old rats and response to 2 min application of 30 μM DA (scale: 1 min, 0.2 mV). Expanded traces (scale: 1 s) show firing in control condition and during DA. The plot shows the firing reduction (% of control) after DA (23 cells from 8 WT, 34 cells from 10 Syn; Mann–Whitney test ***$P < 1.00 \times 10^{-4}$). **b** DA currents (30 μM, 2 min) during voltage clamp (−60 mV; scale: 2 min, 20 pA) and mean (±s.e.m.) dose–response curves (18 cells from 8 WT, 18 cells from 7 Syn; two-way ANOVA: genotype × concentration, $F_{3,29} = 0.66$, $P = 0.586$; concentration, $F_{3,29} = 6.68$, $P = 0.001$; genotype, $F_{1,29} = 8.63$, P = 0.006; WT vs Syn: 1 μM, $P > 0.999$, 3 μM $P = 0.175$, 10 μM, **$P = 0.007$, 30 μM *$P = 0.012$ with Bonferroni's). **c** Baclofen-mediated currents (10 μM, 3 min; −60 mV; scale: 2 min, 50 pA), and mean (±s.e.m.) current amplitude (11 WT cells, 13 Syn cells, 3 rats each; two-way ANOVA: genotype × concentration, $F_{1,20} = 4.78$, $P = 0.041$; concentration, $F_{1,20} = 23.96$, $P < 1.00 \times 10^{-4}$; genotype, $F_{1,20} = 16.04$, $P = 7.00 \times 10^{-4}$. WT vs Syn: 1 μM, $P = 0.337$, 10 μM, **$P = 0.001$ with Bonferroni's). **d** Infrared videomicroscopy of a patched DA neuron and a Fura-2 ratiometric image (scale: 10 μm) showing variations in cytosolic [Ca$^{2+}$]. The plot shows cytoplasmic [Ca$^{2+}$] in DA neurons at −60 mV (23 WT, 26 Syn cells, 7 rats each; ***$P < 1.00 \times 10^{-4}$, Welch's $t$-test). **e** Somatic Ca$^{2+}$ at −60 mV in control conditions and in the presence of Ca$^{2+}$-free bath solution (scale: 2 min). The plot shows the level of [Ca$^{2+}$] reduction (11 WT neurons from 4 rats, 11 Syn neurons from 5 rats; $P = 0.338$ with Welch's $t$-test). **f** [Ca$^{2+}$] levels in DA neurons from slices incubated with isradipine (200 nM, 1 h; 10 WT, 6 Syn neurons, 3 rats each; ***$P = 1.00 \times 10^{-4}$ with Welch's $t$-test. **g** Transient rise in fluorescence ratio following CPA application (10 μM; scale: 2 min) and plot showing smaller CPA-induced Ca$^{2+}$ increase in Syn DA neurons (5 WT, 9 Syn neurons, 3 rats each; ***$P = 0.001$ with Mann–Whitney test). Source data are provided as a Source Data file.

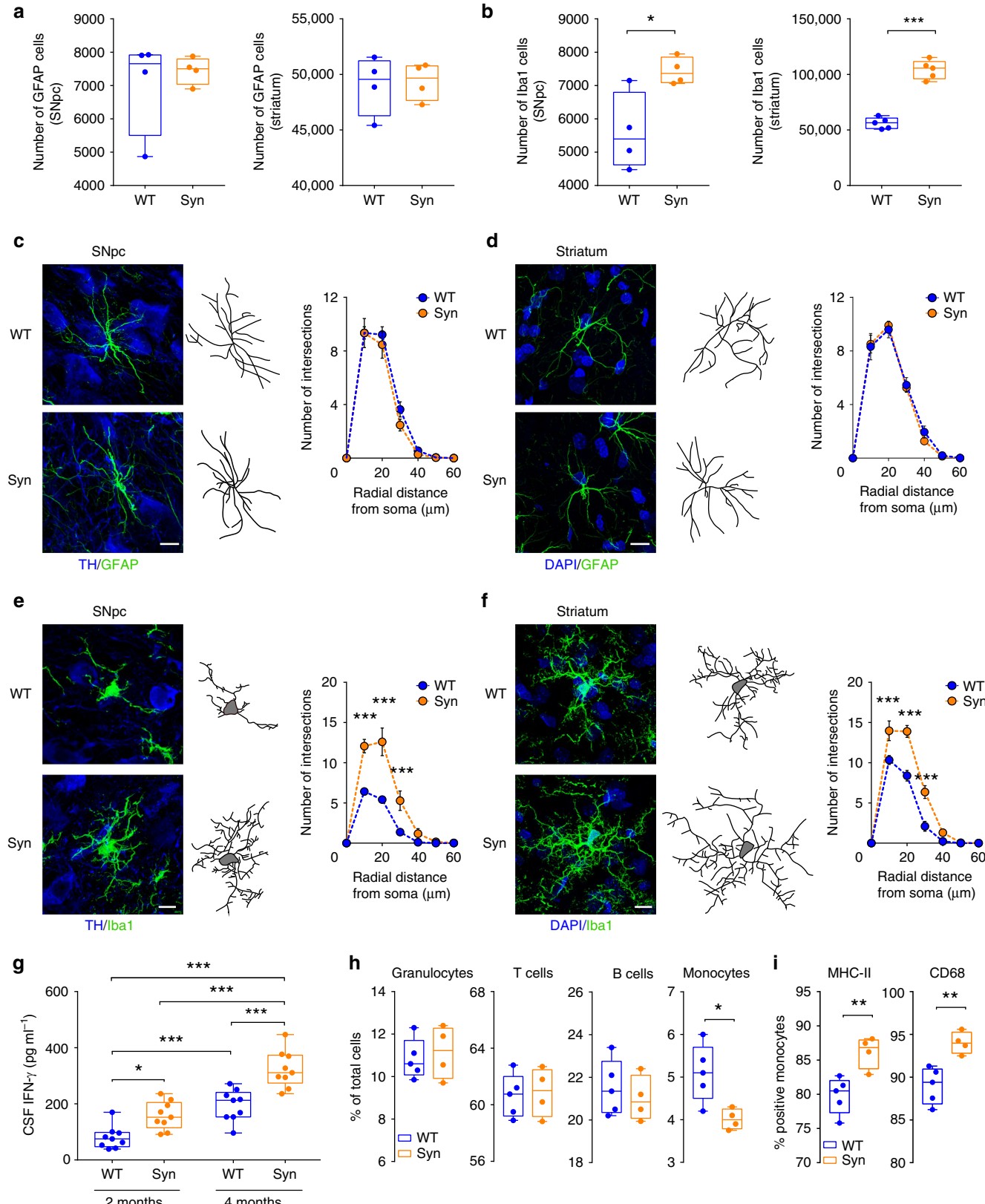

higher expression of the activation markers MHC-II and CD68 (Fig. 4i), suggesting a hyperactive, pro-inflammatory profile.

**α-syn aggregation in Syn rats**. Given that α-syn load induces microglial activation or proliferation in PD[16,17,42,43,47], we next

sought to analyse whether the region-specific inflammation in Syn rats (microglia changes in the SNpc, striatum and hippocampus, but not in the pontine nuclei) could be correlated with selective local aggregation of α-syn species. We used an antibody against conformation-specific α-syn aggregates and fibrils (MJFR-14-6-4-2) to estimate the amount of pathological α-syn in

**Fig. 4** Early neuroinflammatory responses in 4-month-old Syn rats. **a** GFAP$^+$ cell numbers in SNpc and striatum (SNpc: 4 rats each, Mann-Whitney test $P = 0.886$; striatum: 4 rats each, Welch's $t$-test $P = 0.837$). **b** Iba1$^+$ cell numbers in SNpc and striatum (SNpc: 4 rats each, Welch's $t$-test *$P = 0.044$; striatum: 5 rats each, ***$P < 1.00 \times 10^{-4}$ with Welch's $t$-test). **c, d** TH/GFAP staining in SNpc (**c**) and DAPI/GFAP staining in striatum (**d**) of 4-month-old rats (scale: 10 μm) and representative images of 3D-reconstructed astrocytes. The plots show number of intersections (±s.e.m.) in rats (**c**: 4 rats each; two-way repeated-measures ANOVA: genotype × radial distance from soma, $F_{6,36} = 0.583$, $P = 0.742$; genotype, $F_{1,6} = 0.621$, $P = 0.461$; distance, $F_{6,36} = 194.8$, $P < 1.00 \times 10^{-4}$; WT vs Syn for 0–60 μm: $P > 0.05$ with Bonferroni's; **d** 4 rats each; two-way repeated-measures ANOVA: genotype×distance, $F_{6,36} = 0.368$, $P = 0.895$; genotype, $F_{1,6} = 0.039$, $P = 0.851$; distance, $F_{6,36} = 253$, $P < 1.00 \times 10^{-4}$; WT vs Syn for 0–60 μm: $P > 0.05$ with Bonferroni's). See also Supplementary Fig. 3. **e, f** TH/Iba1 staining in SNpc (**e**) and DAPI/Iba1 staining in the striatum (**f**) of 4-month-old rats (scale: 10 μm) and 3D-reconstructed microglia. The plots show number of intersections (±s.e.m.) (**e**: 4 rats each; two-way repeated-measures ANOVA: genotype×distance, $F_{6,36} = 16.58$, $P < 1.00 \times 10^{-4}$; genotype, $F_{1,6} = 17.05$, $P = 0.006$; distance, $F_{6,36} = 132.2$, $P < 1 \times 10^{-4}$; 10 μm ***$P < 1 \times 10^{-4}$, 20 μm ***$P < 1 \times 10^{-4}$, 30 μm ***$P < 8 \times 10^{-4}$, with Bonferroni's; **f** 4 rats each; two-way repeated-measures ANOVA: genotype×distance, $F_{6,36} = 11.99$, $P < 1 \times 10^{-4}$; genotype, $F_{1,6} = 24.0$, $P = 0.003$; distance, $F_{6,36} = 257.9$, $P < 1 \times 10^{-4}$; 10 μm ***$P = 1 \times 10^{-4}$, 20 μm ***$P < 1 \times 10^{-4}$, 30 μm ***$P < 1 \times 10^{-4}$, with Bonferroni's). See also Supplementary Fig. 4. **g** CSF IFN-γ levels in WT and Syn rats at the indicated ages (9 WT and 9 Syn rats per age; two-way ANOVA: genotype × age, $F_{1,32} = 1.95$, $P = 0.172$; genotype, $F_{1,32} = 33.44$, $P < 1.00 \times 10^{-4}$; age, $F_{1,32} = 60.84$, $P < 1.00 \times 10^{-4}$; WT 2-Syn 2 *$P = 0.024$, WT 2-WT 4 ***$P = 5.00 \times 10^{-4}$, WT 2-Syn 4 ***$P < 1.00 \times 10^{-4}$, Syn 2-Syn 4 ***$P < 1.00 \times 10^{-4}$, WT 4-Syn 4 ***$P < 1.00 \times 10^{-4}$ with Bonferroni's). **h** Percentage of peripheral blood cells gated on live leukocytes or on CD11b$^+$ monocytes (5 WT, 4 Syn; granulocytes: $P = 0.696$; T-lymphocytes: $P = 0.827$; B-lymphocytes: $P = 0.544$; monocytes: *$P = 0.019$, with Welch's $t$-test). **i** Percentage of peripheral monocytes expressing MHC-II and CD68 (5 WT, 4 Syn; MHC-II: **$P = 0.007$; CD68: **$P = 0.004$, with Welch's $t$-test). Source data are provided as a Source Data file

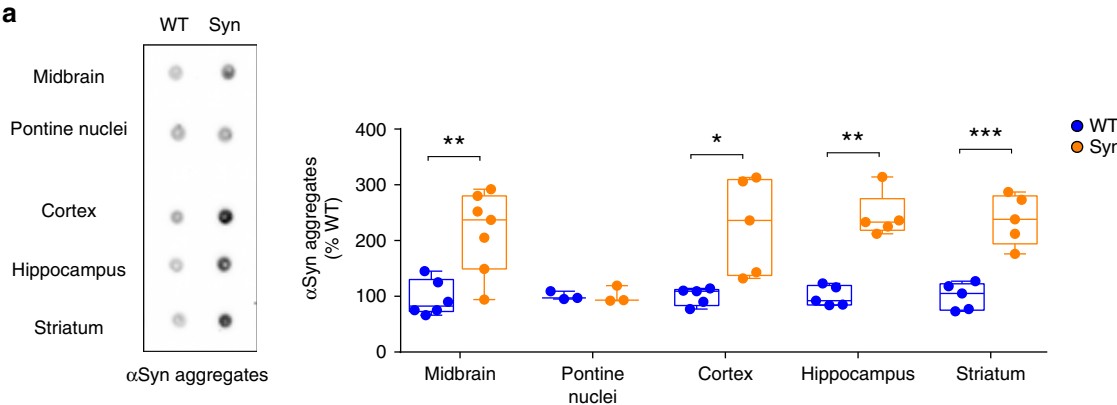

**Fig. 5** Region-specific aggregation of pathological α-syn in Syn rats. **a** Representative dot blot of midbrain, pontine nuclei, cortex, hippocampus and striatum homogenates from 4-month-old WT and Syn rats, immunostained with the conformation-specific α-syn antibody [MJFR-14-6-4-2], and densitometric quantification. The plot shows the levels of α-syn aggregates (expressed as % of WT). (Midbrain: 6 WT rats and 7 Syn rats, Welch's $t$-test **$P = 0.004$; Pontine nuclei: 3 rats per genotype, Welch's $t$-test $P = 0.926$; Cortex: 5 rats per genotype, Welch's $t$-test *$P = 0.030$; Hippocampus: 5 rats per genotype, Mann–Whitney test **$P = 0.008$; Striatum: 5 rats per genotype, Welch's $t$-test ***$P = 0.001$). The full blot and raw data are provided as a Source Data file

homogenates from midbrain, striatum, hippocampus, cortex and pontine nuclei from 4-month-old animals (Fig. 5a). Pathological α-syn accumulation was higher in all the examined areas in Syn rats, apart from the pontine nuclei, a symmetrical picture to the region-specific microglia response, suggesting that the local pathological accumulation of α-syn might trigger the microglia changes in our animal model.

**Pro-resolution deficits in Syn rats**. We next questioned whether the early neuroinflammatory changes in Syn rats were also associated with alterations in the resolution of inflammation, by assessing the CSF and plasma levels of two major resolvins, resolvin D1 (RvD1) and D2 (RvD2) (Fig. 6a). Four-month-old Syn rats displayed higher CSF levels of RvD1 (left in Fig. 6b), but not RvD2 (left in Fig. 6c), compared to aged-matched controls. This increase was age-dependent since RvD1 levels were also higher than 2-month-old Syn rats (left in Fig. 6b). Interestingly, the RvD1 increase in the CSF of 4-month-old Syn rats was coupled to marked reduction in the plasma (right in Fig. 6b). Plasma levels of RvD2 showed no difference compared to WT, despite an age-dependent increase (right in Fig. 6c). To better understand how the resolvin changes in Syn rats are correlated with the disease progression, we also measured RvD1 and RvD2

in 18-month-old rats, showing pronounced α-syn aggregation, marked DA neuron degeneration, striatal denervation and strong motor impairments[27]. RvD1 levels in the CSF and plasma of 18-month-old Syn rats were much lower compared to aged-matched WT (Fig. 6d; compare also with Fig. 6b). This time-course of RvD1 in the CSF (showing normal levels at 2 months, increased levels at 4 months and low levels in aged animals) suggests that the boost in 4-month-old animals might account for a first attempt to counteract the inflammation at its initial stages. Contrary to RvD1, no differences were detected in RvD2 CSF or plasma levels in 18-month-old rats (Fig. 6e).

**RvD1 prevents inflammation, neuronal and motor deficits**. An efficient pro-resolution mechanism is critical for tissue homeostasis and for preventing chronic inflammation[24–26]. Thus, we hypothesized that by chronically treating Syn animals with RvD1 (see Fig. 7a for treatment protocol) might limit the observed deficits and prevent the onset of PD-like features in 4-month-old rats. The dose and modality of the in-vivo RvD1 administration was chosen according to literature[48,49]. Nonetheless, owing to possible interference of plasma proteins and lipid metabolism, in a separate animal group we assessed RvD1 bioavailability with time-course analysis of plasma levels and calculation of

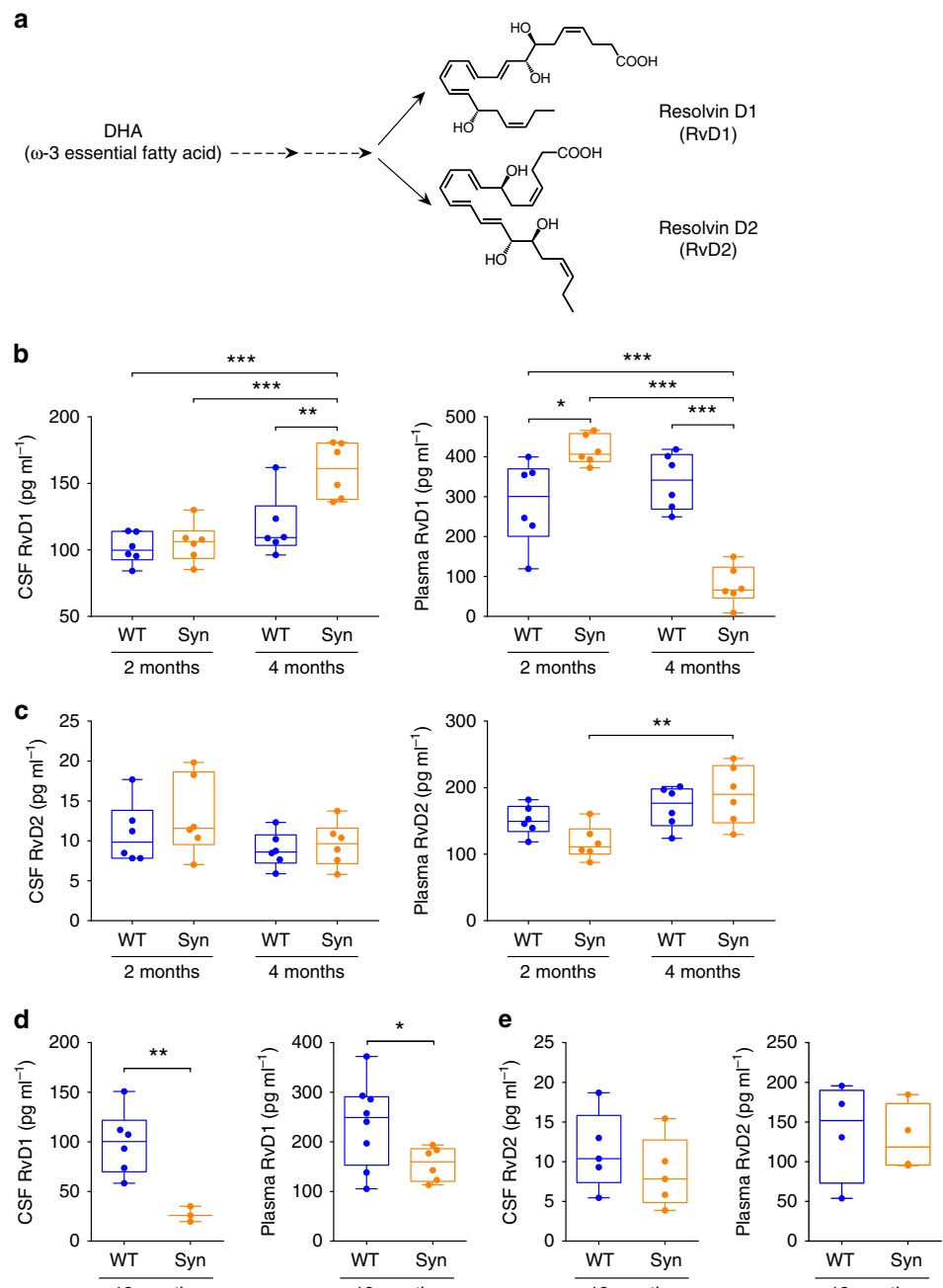

**Fig. 6** Age-dependent changes in central and peripheral RvD1 levels in Syn rats. **a** Chemical structure of RvD1 and RvD2, derived from docosahexaenoic acid (DHA) metabolism. **b** Quantification with ELISA of RvD1 in the CSF (left) and plasma (right) of WT and Syn rats at 2 and 4 months of age (CSF: 6 WT, 6 Syn rats per age; two-way ANOVA: genotype × age, $F_{1,20} = 6.38$, $P = 0.020$; genotype, $F_{1,20} = 9.61$, $P = 0.006$; age, $F_{1,20} = 22.36$, $P = 1.00 \times 10^{-4}$; WT 2 -Syn 4 ***$P = 1.00 \times 10^{-4}$, Syn 2 -Syn 4 ***$P = 3.00 \times 10^{-4}$, WT 4-Syn 4 **$P = 0.004$ with Bonferroni's; plasma: 6 WT, 6 Syn for each age; two-way ANOVA: genotype × age, $F_{1,20} = 46.44$, $P < 1.00 \times 10^{-4}$; genotype, $F_{1,20} = 4.99$, $P = 0.037$; age, $F_{1,20} = 24.64$, $P < 1.00 \times 10^{-4}$; WT 2-Syn 2 *$P = 0.025$, WT 2-Syn 4 ***$P = 3.00 \times 10^{-4}$, Syn 2-Syn 4 ***$P < 1.00 \times 10^{-4}$, WT 4 -Syn 4 ***$P < 1.00 \times 10^{-4}$ with Bonferroni's). **c** ELISA quantifications of CSF and plasma RvD2 from 2- and 4-month-old rats (6 WT and 6 Syn per age; two-way ANOVA for CSF: genotype × age, $F_{1,20} = 0.26$, $P = 0.613$; genotype, $F_{1,20} = 0.94$, $P = 0.343$; age, $F_{1,20} = 3.66$, $P = 0.070$; two-way ANOVA for plasma: genotype × age, $F_{1,20} = 4.03$, $P = 0.059$; genotype, $F_{1,20} = 0.339$, $P = 0.567$; age, $F_{1,20} = 12.43$, $P = 0.002$; Syn 2-Syn 4 **$P = 0.005$ with Bonferroni's). **d**, **e** CSF and plasma RvD1 (**d**) and RvD2 (**e**) from 18-month-old rats (**d**: CSF: 6 WT and 3 Syn rats, **$P = 0.002$; plasma: 8 WT and 6 Syn rats, *$P = 0.039$ with Welch's $t$-test; **e**: CSF: 5 rats per genotype, $P = 0.379$; plasma: 4 per genotype, $P = 0.818$ with Welch's $t$-test). Source data are provided as a Source Data file

pharmacokinetics parameters. After a single injection (i.p., 0.2 μg kg$^{-1}$) the plasma concentration of RvD1 peaked at 1 h, stayed almost constant at 3 h and returned to baseline after 36 h (Supplementary Fig. 5b), indicating rapid distribution into the bloodstream and elimination from the vascular compartment due to metabolism and/or diffusion into the blood–brain barrier.

We then examined the effects of the 2-month RvD1 treatment on microglia activation and morphology, IFN-γ levels, DA neuron properties, nigrostriatal transmission and behaviour. Syn rats treated with RvD1 showed reduced Iba1$^+$ microglia numbers compared to saline-treated animals both in the SNpc (Fig. 7b) and striatum (Fig. 7c), whereas RvD1 had no effect on WT rats.

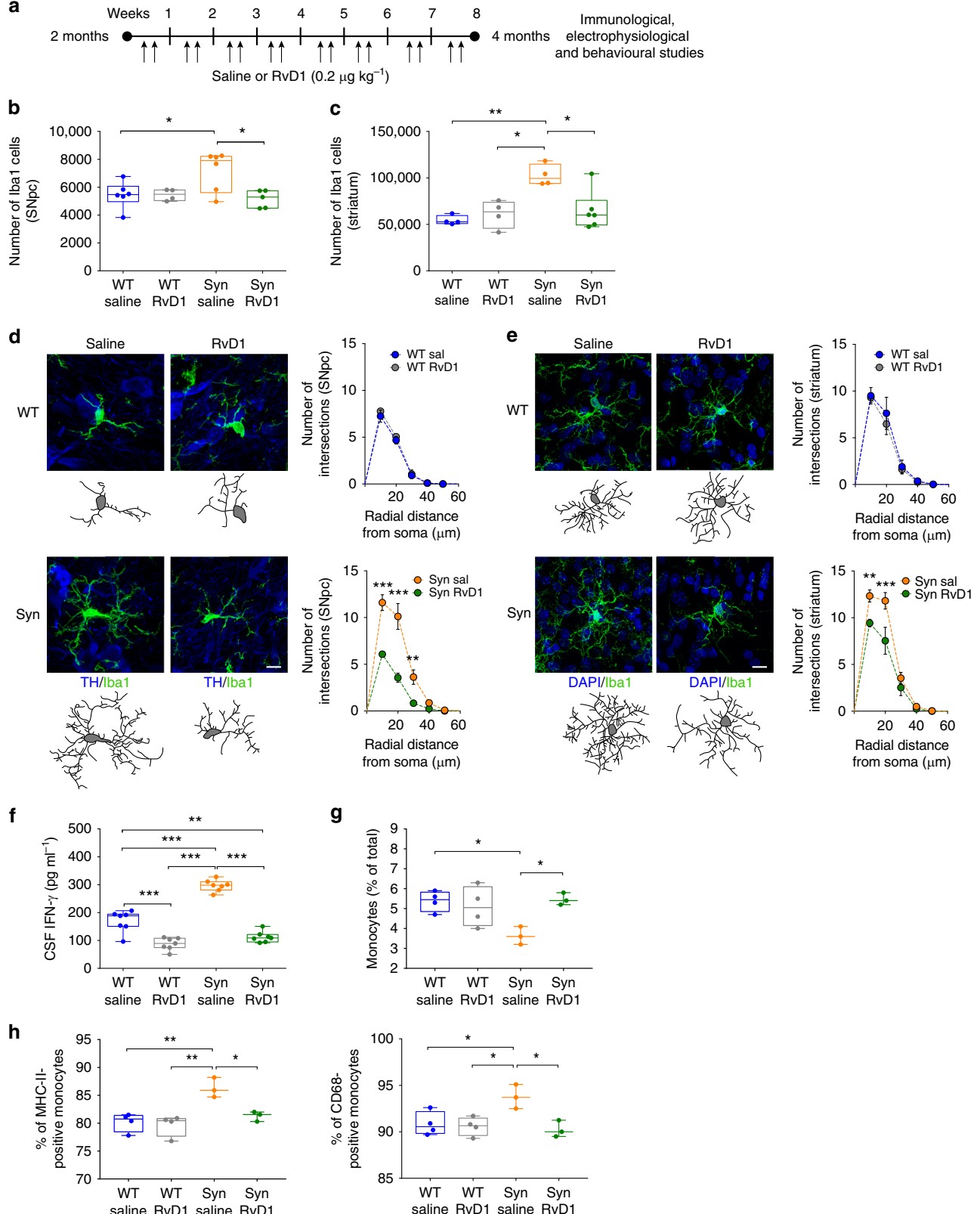

RvD1 treatment could not restore Iba1$^+$ numbers in the dorsal hippocampus (Supplementary Fig. 8), yet in this area, as well as in the SNpc and striatum, the microglia of RvD1-treated Syn rats showed lower numbers of intersections, nodes and endings and reduced length of processes compared to saline-treated animals (Fig. 7d, e and Supplementary Figs. 7 and 8). Additionally, RvD1

treatment significantly reduced the CSF levels of IFN-γ in both Syn and WT rats, cancelling the difference between genotypes (Fig. 7f, compare with Fig. 4g). These data are supported by the fact that, following the 2-month treatment, the RvD1 concentration was not only increased in the plasma but also in the CSF of RvD1-treated rats (Supplementary Fig. 5c, d), indicating its ability

**Fig. 7** RvD1 resolves neuroinflammation in 4-month-old Syn rats. **a** RvD1 treatment scheme ($0.2\,\mu g\,kg^{-1}$). Injections were performed twice a week for 2 months, starting from 2- and until 4 months of age. **b**, **c** Iba1$^+$ numbers in SNpc (**b**) and striatum (**c**). (**b**: 6 WT/saline, 4 WT/RvD1; 6 Syn/saline, 4 Syn/RvD1; ANOVA: genotype × treatment, $F_{1,17} = 5.25$, $P = 0.035$; treatment, $F_{1,17} = 5.33$, $P = 0.034$; genotype, $F_{1,17} = 2.69$, $P = 0.119$; WT/saline-Syn/saline *$P = 0.047$, Syn/saline-Syn/RvD1 *$P = 0.022$ with Bonferroni's; **c**: 4 WT/saline, 4 WT/RvD1; 4 Syn/saline, 6 Syn/RvD1; ANOVA: genotype × treatment, $F_{1,14} = 9.36$, $P = 0.009$; treatment, $F_{1,14} = 4.59$, $P = 0.050$; genotype, $F_{1,14} = 12.72$, $P = 0.003$; WT/saline-Syn/saline **$P = 0.003$, WT/RvD1-Syn/saline *$P = 0.010$, Syn/saline-Syn/RvD1 *$P = 0.010$ with Bonferroni's). **d**, **e** TH/Iba1 staining in SNpc (**d**) and DAPI/Iba1 staining in striatum (**e**; scale, $10\,\mu m$) and 3D-reconstructed microglia. Number of intersections (±s.e.m.) in SNpc (**d**: 4 rats each; repeated-measures ANOVA for WT: treatment×distance, $F_{6,36} = 0.36$, $P = 0.898$; treatment, $F_{1,6} = 0.71$, $P = 0.431$; distance, $F_{6,36} = 283.9$, $P < 1.00 \times 10^{-4}$; 0–60 $\mu m$ $P > 0.05$ with Bonferroni's; for Syn: treatment × distance, $F_{6,36} = 18.9$, $P < 1.00 \times 10^{-4}$; treatment, $F_{1,6} = 27.1$, $P = 0.002$; distance, $F_{6,36} = 130.7$, $P < 1.00 \times 10^{-4}$; $10\,\mu m$ ***$P < 1.00 \times 10^{-4}$, $20\,\mu m$ ***$P < 1.00 \times 10^{-4}$, $30\,\mu m$ **$P = 0.003$, with Bonferroni's) and striatum (**e**: 4 rats each; repeated-measures ANOVA for WT: treatment × distance, $F_{6,36} = 0.26$, $P = 0.951$; treatment, $F_{1,6} = 0.21$, $P = 0.667$; distance, $F_{6,36} = 94.82$, $P < 1.00 \times 10^{-4}$; 0–60 $\mu m$ $P > 0.05$ with Bonferroni's; for Syn: treatment× distance, $F_{6,36} = 5.72$, $P = 3.00 \times 10^{-4}$; treatment, $F_{1,6} = 6.94$, $P = 0.039$; distance, $F_{6,36} = 180.9$, $P < 1.00 \times 10^{-4}$; $10\,\mu m$ **$P = 0.006$, $20\,\mu m$ ***$P < 1.00 \times 10^{-4}$, with Bonferroni's). See also Supplementary Fig. 7. **f** CSF IFN-γ levels (7 rats each; ANOVA: genotype×treatment, $F_{1,24} = 26.81$, $P < 1.00 \times 10^{-4}$; treatment, $F_{1,24} = 177.2$, $P < 1.00 \times 10^{-4}$; genotype, $F_{1,24} = 58.79$, $P < 1.00 \times 10^{-4}$; WT/saline-WT/RvD1 ***$P < 1.00 \times 10^{-4}$, WT/saline-Syn/saline ***$P < 1.00 \times 10^{-4}$, WT/saline-Syn/RvD1 **$P = 0.003$, WT/RvD1-Syn/saline ***$P < 1.00 \times 10^{-4}$, Syn/saline-Syn/RvD1 ***$P < 1.00 \times 10^{-4}$ with Bonferroni's). **g** Monocyte (% of total blood cells) following treatment (4 WT/saline, 4 WT/RvD1; 3 Syn/saline, 3 Syn/RvD1; ANOVA: genotype×treatment, $F_{1,10} = 8.58$, $P = 0.015$; genotype, $F_{1,10} = 3.65$, $P = 0.085$; treatment, $F_{1,10} = 4.69$, $P = 0.056$; WT/saline-Syn/saline *$P = 0.039$, Syn/saline-Syn/RvD1 *$P = 0.043$ with Bonferroni's). **h** Percentage of MHC-II and CD68-expressing monocytes (4 WT/saline, 4 WT/RvD1; 3 Syn/saline, 3 Syn/RvD1; MHC-II: ANOVA: genotype × treatment, $F_{1,10} = 6.20$, $P = 0.032$; genotype, $F_{1,10} = 18.69$, $P = 0.002$; treatment, $F_{1,10} = 9.653$, $P = 0.011$; WT/saline-Syn/saline **$P = 0.004$, WT/RvD1-Syn/saline **$P = 0.002$, Syn/saline-Syn/RvD1 *$P = 0.025$ with Bonferroni's; CD68: ANOVA for genotype × treatment, $F_{1,10} = 7.05$, $P = 0.024$; genotype, $F_{1,10} = 4.51$, $P = 0.060$; treatment, $F_{1,10} = 9.651$, $P = 0.011$; WT/saline-Syn/saline *$P = 0.042$, WT/RvD1-Syn/saline *$P = 0.025$, Syn/saline-Syn/RvD1 *$P = 0.021$ with Bonferroni's). Source data are provided as a Source Data file

to cross the blood–brain barrier. Of note, the treatment restored peripheral monocyte levels in Syn rats (Fig. 7g), without affecting the numbers of other cell populations (Supplementary Fig. 6b), and reduced the expression of MHC-II and CD68 monocyte activation markers (Fig. 7h), indicating that RvD1 can impact on peripheral immune responses.

We next tested whether RvD1 could also prevent the functional alterations in Syn animals. Unlike saline-treated Syn rats, RvD1-treated animals showed normal DA neuron firing (Fig. 8a), normal autoreceptor function following bath-applied DA (Fig. 8b), reduced cytoplasmic Ca$^{2+}$ levels (Fig. 8c) and improved evoked DA release in striatal slices (Fig. 8d). These effects were paralleled by the prevention of appearance of motor deficits. Indeed, RvD1-treated Syn rats showed normal crossing into the centre zone during an open field test (Fig. 8e) and improved performance on the accelerating rotarod compared to saline-treated Syn animals (Fig. 8f), overall indicating that the potentiation of the RvD1 pathway can efficiently prevent the neurophysiological and motor deficits in Syn rats.

**Decreased endogenous RvD1 in early-PD patients.** Since animal models reproduce only partially the complexity of the human pathology, to obtain a relevance of our results in humans we sought to investigate whether early-PD patients present signs of inflammation and changes in the pro-resolution pathway. Thus, we analysed the levels of different cytokines, and of RvD1 and RvD2 in the CSF and plasma of control subjects and early-PD patients. Patients were defined as early-PD if they were newly-diagnosed (symptom duration 13 ± 5 months), untreated (de novo) and mildly affected (Hoehn and Yahr, H&Y scale 1.38 ± 0.18; see Table 1 and Supplementary Table 1 for more clinical features). PD patients showed generally higher levels of IFN-γ, TNF-α, IL-4 and IL-10 in CSF, although being significant only for IL-4. Some cytokines such as IL-1β, IL-6 and TNF-α could not be detected in any control subject (Fig. 9a). On the other hand, all cytokines were detected in the plasma of PD patients and, overall, PD patients displayed an increase in most of them, with IFN-γ and IL-10 showing a significant variation compared to control subjects (Fig. 9b). Importantly, RvD1 levels in both the CSF and plasma were dramatically reduced in PD patients compared

to control subjects (Fig. 9c), while RvD2 levels were unchanged (Fig. 9d).

## Discussion

We report here that 4-month-old Syn rats—overexpressing the human α-syn—are characterized by functional alterations in SNpc DA neurons, reduced striatal DA outflow and subtle motor deficits. These changes are age-dependent and precede DA neuron degeneration, striatal denervation or severe motor impairment. Importantly, they are associated with early neuroinflammation and alterations of RvD1, a lipid mediator responsible for resolution of inflammation. RvD1-based impairments were also seen in early-PD patients. Chronic treatment of Syn animals with RvD1, starting from a stage preceding the observed alterations, prevents the onset of PD by attenuating neuroinflammation.

Our data confirm the hypothesis that human α-syn overexpression leads to multiple functional alterations that can contribute to SNpc DA neuron vulnerability and eventually lead to progressive degeneration and overt motor deficits in older Syn animals[27] and in humans[1]. These functional deficits include reduced neuronal excitability, alterations in K$^+$ conductances and increased apamin-sensitive AHC linked to higher somatic [Ca$^{2+}$][37–40] at a holding potential that prevented somatic spike firing. Given the appealing hypothesis that links altered Ca$^{2+}$ homeostasis with mitochondrial dysfunction and increased susceptibility of DA neurons in PD[11,41,50], our data are of particular relevance. At −60 mV we could not detect effects of isradipine on the cytoplasmic [Ca$^{2+}$] but this could be due to the fact that L-type channels are mostly shut at near-resting conditions[51]. Of note, our technique, combining somatic patch-clamp with microfluorometry, does not permit to detect dendritic [Ca$^{2+}$] changes via L-type and other VGCCs, but this does not exclude that Ca$^{2+}$ entry from dendritic VGCCs during pacemaking[52,53] might contribute to DA neuron dysfunction in Syn rats. Nonetheless, our experiments highlight a reduced storing capacity of the CPA-sensitive ER, likely being the reason for the high cytoplasmic [Ca$^{2+}$] in 4-month-old Syn DA neurons. The reduced storing capacity in Syn rats could be due to reduced SERCA functioning and/or due to increased ER Ca$^{2+}$ channel opening, allowing Ca$^{2+}$ to flow back to the cytoplasm. A more detailed

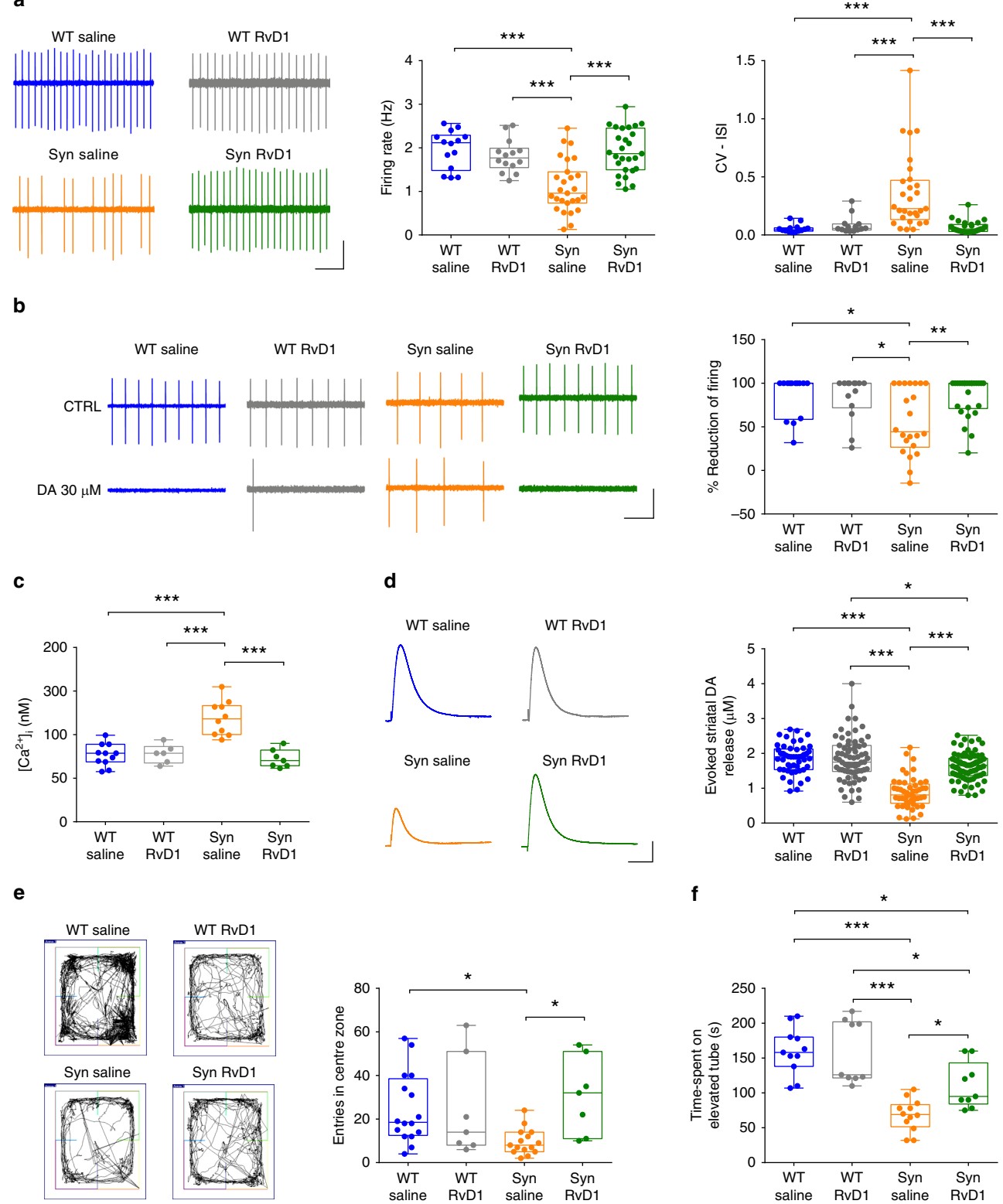

investigation of the interplay between α-syn, Ca²⁺ buffering and mitochondrial function in these animals would be of great interest, particularly since α-syn can interfere with SERCA function, resulting in dysregulation of cytosolic Ca²⁺ levels[54].

The deficits in Syn rats appear at 2–4 months of age, meaning that this time-window is ideal for studying the earliest effects of α-syn overexpression. In fact, unlike many classic toxin-based PD models that fail to effectively mimic the entirety of the human

disease[55], the Syn model is particularly promising as it recapitulates most PD features[27,28], including early inflammation. Indeed, our study extends previous findings on a link between α-syn, microglia activation and cytokine production in other PD models[42–47], showing also that these processes occur at early disease stages and are linked to alterations in the RvD1-mediated mechanism of resolution of inflammation. This is in line with evidence that α-syn-overexpressing neurons show altered

**Fig. 8** RvD1 treatment prevents neuronal and motor deficits in 4-month-old Syn rats. **a** DA neuron firing in treated rats (scale: 2 s, 0.2 mV) and firing frequency plots (14 WT/saline, 14 WT/RvD1 cells from 3 rats; 27 Syn/saline and 27 Syn/RvD1 cells from 4 rats; ANOVA for genotype × treatment, $F_{1,78} = 16.34$, $P = 1.00 \times 10^{-4}$; genotype, $F_{1,78} = 10.04$, $P = 0.002$; treatment, $F_{1,78} = 7.43$, $P = 0.008$; WT/saline-Syn/saline ***$P < 1.00 \times 10^{-4}$, WT/RvD1-Syn/saline ***$P = 5.00 \times 10^{-4}$, Syn/saline−Syn/RvD1 ***$P < 1.00 \times 10^{-4}$, with Bonferroni's) and CV-ISI (14 WT/saline, 14 WT/RvD1 cells from 3 rats; 29 Syn/saline and 27 Syn/RvD1 cells from 4 rats; ANOVA for genotype × treatment, $F_{1,80} = 12.58$, $P = 7.00 \times 10^{-4}$; genotype, $F_{1,80} = 10.10$, $P = 0.002$; treatment, $F_{1,80} = 8.40$, $P = 0.005$; WT/saline-Syn/saline *** $P < 1.00 \times 10^{-4}$, WT/RvD1-Syn/saline ***$P = 3.00 \times 10^{-4}$, Syn/saline-Syn/RvD1 ***$P < 1.00 \times 10^{-4}$, with Bonferroni's). **b** DA neuron firing (scale: 1 s, 0.2 mV) from treated rats before (CTRL) and during DA (30 µM, 2 min) and plot showing % inhibition by DA (14 WT/saline, 14 WT/RvD1 cells from 3 rats; 22 Syn/saline and 26 Syn/RvD1 cells from 4 rats; ANOVA for genotype × treatment, $F_{1,72} = 5.06$, $P = 0.028$; genotype, $F_{1,72} = 4.38$, $P = 0.040$; treatment, $F_{1,72} = 3.93$, $P = 0.051$; WT/saline-Syn/saline *$P = 0.021$, WT/RvD1-Syn/saline *$P = 0.035$, Syn/saline-Syn/RvD1 **$P = 0.005$, with Bonferroni's). **c** Cytoplasmic $[Ca^{2+}]$ in DA neurons at −60 mV (11 WT/saline, 6 WT/RvD1 cells from 3 rats; 10 Syn/saline, 7 Syn/RvD1 cells from 4 rats; ANOVA: genotype×treatment, $F_{1,30} = 21.5$, $P < 1.00 \times 10^{-4}$; genotype, $F_{1,30} = 12.82$, $P = 0.001$; treatment, $F_{1,30} = 19.08$, $P = 1.00 \times 10^{-4}$; WT/saline-Syn/saline ***$P < 1.00 \times 10^{-4}$, WT/RvD1-Syn/saline ***$P < 1.00 \times 10^{-4}$, Syn/saline-Syn/RvD1 ***$P < 1.00 \times 10^{-4}$, with Bonferroni's). **d** Amperometric traces (scale: 500 ms, 50 pA) and striatal DA release (49 WT/saline, 71 WT/RvD1 slices from 4 rats; 49 Syn/saline, 77 Syn/RvD1 slices from 4 rats; ANOVA: genotype×treatment, $F_{1,242} = 36.43$, $P < 1.00 \times 10^{-4}$; genotype, $F_{1,242} = 90.56$, $P < 1.00 \times 10^{-4}$; treatment, $F_{1,242} = 35.29$, $P < 1.00 \times 10^{-4}$; WT/saline-Syn/saline ***$P < 1.00 \times 10^{-4}$, WT/RvD1-Syn/saline ***$P < 1.00 \times 10^{-4}$, Syn/saline-Syn/RvD1 ***$P < 1.00 \times 10^{-4}$, WT/RvD1-Syn/RvD1 *$P = 0.038$ with Bonferroni's). **e** Entries in centre zone during open field test in treated rats (16 WT/saline, 7 WT/RvD1, 15 Syn/saline and 7 Syn/RvD1 rats; two-way ANOVA for genotype×treatment, $F_{1,41} = 4.75$, $P = 0.035$; genotype, $F_{1,41} = 0.85$, $P = 0.362$; treatment, $F_{1,41} = 4.59$, $P = 0.038$; WT/saline-Syn/saline *$P = 0.048$, Syn/saline-Syn/RvD1 *$P = 0.025$, with Bonferroni's). **f** Time performance in the accelerating rotarod (11 WT/saline, 9 WT/RvD1, 12 Syn/saline, 9 Syn/RvD1 rats; ANOVA: genotype × treatment, $F_{1,37} = 4.34$, $P = 0.044$; genotype, $F_{1,37} = 43.04$, $P < 1.00 \times 10^{-4}$; treatment, $F_{1,37} = 3.64$, $P = 0.064$; WT/saline-Syn/saline ***$P < 1.00 \times 10^{-4}$, WT/saline-Syn/RvD1 *$P = 0.014$, WT/RvD1-Syn/saline ***$P < 1.00 \times 10^{-4}$, WT/RvD1-Syn/RvD1 *$P = 0.030$, Syn/saline-Syn/RvD1 *$P = 0.043$). Source data are provided as a Source Data file

### Table 1 Clinical features of PD patients and control (CTRL) subjects

|  | PD | CTRL |
|---|---|---|
| N | 8 | 8 |
| Age (± s.e.m.) | 62.7 ± 2.6 | 62.6 ± 4.6 |
| Sex (M/F) | 6/2 | 5/3 |
| Symptom duration at time of hospital admission (months ± s.e.m.) | 13 ± 5 | Not applicable |
| MMSE score (±s.e.m.) | 27 ± 0.60 | 27 ± 0.39 |
| UPDRS III score (±s.e.m.) | 24 ± 1.84 | Not applicable |
| H&Y scale (±s.e.m.) | 1.38 ± 0.18 | Not applicable |
| TAU (pg ml$^{-1}$ ± s.e.m.) | 149.63 ± 5.70 | 197 ± 22.22 |
| pTAU (pg ml$^{-1}$ ± s.e.m.) | 39.25 ± 1.58 | 50.00 ± 4.02 |
| $\beta_{40}$ (pg ml$^{-1}$ ± s.e.m.) | 1118 ± 48 | 1137 ± 52 |
| $\beta_{42}$ (pg ml$^{-1}$ ± s.e.m.) | 9338 ± 572 | 9441 ± 374 |
| $\beta_{40}/\beta_{42}$ | 0.12 ± 0.00 | 0.13 ± 0.00 |

See Supplementary Table 1 for additional details

polyunsaturated fatty-acid composition and that this process modifies cell membrane fluidity[56], that can affect the positioning and mobility of membrane proteins and diverse cellular functions such as receptor signalling, transporter function, channel conductance and neurotransmitter release. Indeed, a reduction in dopaminergic synaptic vesicle number and of DA levels can be observed in the frontal cortex of rats fed with a diet deficient in omega-3 fatty acids, especially in docosahexaenoic acid (DHA)[57], the precursor of RvD1 (Fig. 5a).

In our model the microglia response is precocious, region-specific and involves morphological changes. Indeed, the activation of microglia in the midbrain, striatum and hippocampus of Syn rats is strictly related to the pathological accumulation of α-syn fibrils analysed with a conformation-specific antibody. Conversely, microglia remained unchanged/inactivated in the pontine nuclei, a brain area showing lack of α-syn aggregation. These data are in line with many other works showing that pathological α-syn aggregation can trigger microglia activation in PD[42,43,46,47]. However, given that a dot blot cannot discriminate between intracellular or extracellular α-syn aggregation, our data are insufficient to directly answer the question of whether the microglia activation is due to extracellular α-syn aggregates

directly, or due to indirect effects such as neuronal dysfunction following intracellular α-syn aggregation. Of note, although the precise relationship between microglia morphological changes and their pro-inflammatory status is controversial[58], our results indicate that α-syn overexpression induces a more hyper-ramified phenotype that is typically associated with a pro-inflammatory state, characterized by secretion of pro-inflammatory cytokines, nitric oxide and chemokines[59]. Although our experiments do not prove that IFN-γ is directly secreted from microglia, the parallel trend between microglia responses and IFN-γ increase in Syn animals suggests that α-syn overexpression is associated with a microglial pro-inflammatory phenotype. Additionally, while IFN-γ had no effect on WT slices, the combined presence of IFN-γ with the overexpressed α-syn in Syn slices could accelerate the appearance of deficits in striatal DA release at an age when, in the absence of IFN-γ, DA release is normal. It could be that α-syn accumulation is the trigger for IFN-γ–mediated inflammation. However, further studies are needed to understand the precise role of α-syn overexpression in microglia dysregulation.

The reduced numbers and more active state of monocytes in Syn rats, which are among the main producers of resolvins, and their full recovery following RvD1 treatment, suggest a role for these peripheral cells in the pathogenesis of PD. Our data are in line with other studies reporting peripheral inflammation in PD, including changes in patients' blood immune cells[60–62]. Of note, the reduction of peripheral monocytes in Syn rats might explain the increased Iba1$^+$ cell numbers in SNpc, striatum and hippocampus, given that both resident microglia and infiltrated monocyte-derived macrophages can express Iba1, in line with recent evidence of monocyte, T- or B-cell infiltration in α-syn based models[63–66]. Nonetheless, we cannot exclude other possible explanations for blood monocyte reduction in Syn rats, including cell death following inflammatory activation or re-mobilization from primary/secondary lymphoid organs, including bone marrow and spleen, where they are recruited for clearance. Importantly, although no changes were observed in the numbers of other peripheral immune cells, their role cannot be excluded inasmuch as their involvement might entail functional rather than numerical changes. Lack of apparent functional changes in Syn rats could be due to selective variation of specific sub-populations or simply due to the fact that they were analysed at an early disease stage, when changes might not have occurred yet.

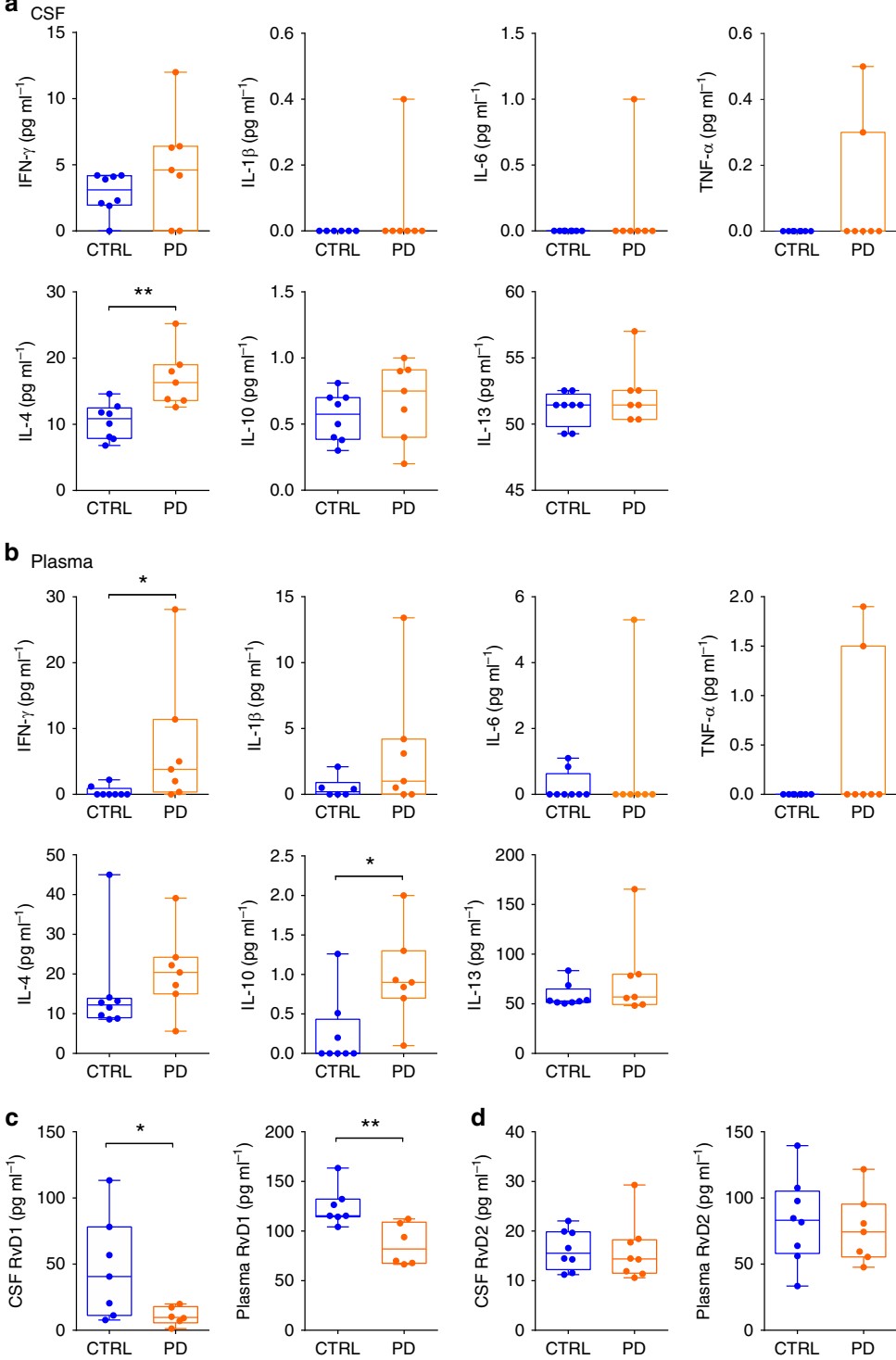

**Fig. 9** CSF and plasma cytokines and resolvins in PD patients and control subjects. **a** Quantification with ELISA of CSF levels of pro- and anti-inflammatory cytokines in control subjects (CTRL) and PD patients (IFN-γ: 8 CTRL and 7 PD, $P = 0.183$; IL-1β: 6 CTRL and 7 PD, $P > 0.999$; IL-6: 8 CTRL and 7 PD, $P = 0.467$; TNF-α: 8 CTRL and 7 PD, $P = 0.200$; IL-4: 8 CTRL and 7 PD, **$P = 0.004$; IL-10: 8 CTRL and 7 PD, $P = 0.294$; IL-13: 8 CTRL and 7 PD, $P = 0.436$, all with Mann–Whitney test). **b** Quantification with ELISA of plasma levels of cytokines in CTRL and PD patients (IFN-γ: 8 CTRL and 7 PD, *$P = 0.013$; IL-1β: 6 CTRL and 7 PD, $P = 0.169$; IL-6: 8 CTRL and 7 PD, $P > 0.999$; TNF-α: 8 CTRL and 7 PD, $P = 0.200$; IL-4: 8 CTRL and 7 PD, $P = 0.121$; IL-10: 8 CTRL and 7 PD, *$P = 0.013$; IL-13: 8 CTRL and 7 PD, $P = 0.515$, all with Mann–Whitney test). **c** Quantification of CSF and plasma levels of RvD1 in CTRL and PD patients (7 CTRL and 6 PD patients; CSF: *$P = 0.049$; Plasma: **$P = 0.007$, with two-tailed Welch's $t$-test). **d** Quantification of CSF and plasma levels of RvD2 in PD and CTRL patients. No differences were detected between the two groups (CSF: 8 CTRL and 8 PD patients; $P = 0.936$; Plasma: 8 CTRL and 7 PD patients $P = 0.667$, with two-tailed Welch's $t$-test). Source data are provided as a Source Data file

To conclude, the fact that RvD1 changes and immune cell variations occur both in the blood and in the brain suggests that they could be the result of synergistic peripheral and central immune responses, or a yet-to-be demonstrated effect-and-cause relationship between the two phenomena.

Although candidate drugs targeting inflammation in PD can attenuate behaviour deficits and/or DA neuron loss in animal models, in clinical studies they only show moderate effects[47]. In this context, our observation that RvD1 is altered in both Syn rats and PD patients pinpoint the RvD1-mediated pathway as a novel candidate to be investigated. The time-course of changes in RvD1 CSF levels in Syn rats (normal levels at 2 months, increased levels at 4 months and low levels in aged animals) suggests that the RvD1 boost in the CSF of 4-month-old rats could reflect an attempt to fight inflammation at its initial stages, being a molecule of early intervention against inflammation[24–26]. Our experiments do not permit to discern why RvD1 levels drop so drastically in older animals. This might be due to α-syn overexpression interfering with RvD1 synthesis. For instance, α-syn can interact with DHA and other polyunsaturated fatty acids, dynamically altering their structural conformation[67–69]. Alternatively, the drop of RvD1 might be due to overall failure of the immune system to chronically sustain high RvD1 levels. Nonetheless, the reduction in plasma RvD1 levels led us to assume that an early-starting administration of RvD1 could resolve the neuroinflammatory and behavioural alterations in 4-month-old rats. Interestingly, RvD1 not only prevented changes in IFN-γ levels, microglia activation and peripheral monocyte levels, but also prevented functional deficits and motor impairments. The ability of RvD1 to prevent α-syn-induced deficits is in line with the only two other studies available in the literature, reporting a beneficial effect of RvD1 and RvD2 in inhibiting inflammation and behavioural effects in an MPP⁺-induced in vitro PD model and in a lipopolysaccharide-induced rat model of PD, respectively[70,71].

The RvD1 effects are likely mediated via the ALX/FPR2 receptor, expressed in both glial cells and neurons. This receptor's expression is heterogeneous and varies according to cell type, brain area and even the pathologic state of the brain tissue[26,49,72,73], suggesting that both neurons and microglia could mediate the protective effects of RvD1 either directly or indirectly via release of soluble factors. Our previous finding of a RvD1-mediated mechanism for halting neuroinflammation via activation of ALX/FPR2 receptor-regulated miRNAs[49] suggests that this circuit might also be involved in Syn rats. Although further studies are needed to unravel such mechanisms and to evaluate the long-term role of RvD1 in Syn rats, by examining its efficacy at an older age when degeneration is detectable[27], our study suggests that early potentiation of the pro-resolving pathway might represent a novel disease-modifying treatment, able to delay or prevent neurophysiological and behavioural deficits.

Of note, the evidence that PD patients show peripheral and central reduction in RvD1 levels allows to make a valuable comparison between a common dysregulated inflammatory pathway in rats and humans, where only RvD1 is altered in both species. Indeed, the reduction of RvD1 in the blood and CSF of PD patients is an almost symmetrical picture to what we observed in 18-month-old Syn rats, but differs from the early increase of CSF RvD1 observed in 4-month-old Syn animals. This highlights the usefulness of the rat model, that allows for characterization of earlier changes in RvD1 levels that would not be possible in patients. Nonetheless, due to the complexity of the human disease, direct comparison of data from humans and rats should be done cautiously. Importantly, the finding of RvD1 reduction in PD patients strengthens our findings in the animal model, providing a translational significance. To the best of our knowledge, ours is the first evidence of impairment in a specific pro-resolving

mediator in PD patients, that supports the notion that neuroinflammation in PD could be a consequence of disruption of the resolution process. Indeed, other conditions characterized by persistent or unresolved inflammation, including atherosclerosis, chronic obstructive pulmonary disease, diabetes, rheumatoid arthritis, Alzheimer's and amyotrophic lateral sclerosis, are associated with altered metabolism and function of pro-resolving mediators[26].

In conclusion, our study provides further proof of the critical involvement of inflammation in PD but also sets the basis for using RvD1 as a clinical biomarker of inflammation, also highlighting the translational potential of endogenous pro-resolving mediators.

## Methods

**Animals and pharmacological treatment.** Male homozygous BAC transgenic rats (Sprague-Dawley background) overexpressing the full-length human SNCA locus under the control of the endogenous human regulatory elements (Syn rats)[27] and WT Sprague-Dawley were used at different ages, as specified in the text. Experimental animals were obtained by crossing heterozygous males with heterozygous female rats, and were confirmed as WT or Syn following genotyping using quantitative PCR using DNA from ear biopsies and the primers for copy numbers of the α-syn transgene: SynProm-F: 5′-ccgctcgagcggtaggaccgcttgttttagac-3′ and LC-SynPromR: 5′-cctctttc cacgccactatc-3′, normalized to the rat β-actin reference gene with primers: β-actin-F: 5′-agccatgtacgtagccatcca-3′ and β-actin-R: 5′-tctccggagtccatcacaatg-3′.

All experiments were carried out in line with the ethical guidelines of the European Council Directive (2010/63/EU) and experimental approval was obtained from the Italian Ministry of Health (protocol #528/2017PR).

For in vivo pharmacological treatment, animals were injected intraperitoneally (i.p.) with either 17(S)-Resolvin D1 (RvD1; Cayman Chemicals; 0.2 µg kg⁻¹) dissolved in ethanol, or with 0.9% saline alone, twice a week for 8 consecutive weeks, starting at 2 months of age. For RvD1 pharmacokinetics analysis, rats were injected with a single dose of RvD1 (0.2 µg kg⁻¹) and plasma was collected at different time points post injection (0, 1, 3, 6, 24 and 36 h). RvD1 concentration was measured as described below and different pharmacokinetics parameters were calculated from the obtained Pearson's correlation logarithmic curve.

**Open field test.** General motor activity (horizontal and vertical locomotor activity) was evaluated in an open field (60 × 60 cm arena with a black floor and transparent Plexiglass walls). Animals were habituated for 1 h to the testing room before the beginning of the test. During the test, each animal was placed in the centre of the arena and freely allowed to explore the apparatus for 10 min. Experiments were recorded with a video camera suspended above the arena and data were analysed with Smart PanLab (Harvard Apparatus). The parameters we evaluated were the distance travelled and the entries in the central zone. The open field arena was cleaned with 70% ethanol between sessions.

**Rotarod test.** Motor coordination and balance were tested using an accelerating Rotarod (TSE Systems GmbH, Germany). The apparatus consisted of a rod suspended horizontally at a height of 14 cm from the floor. During training, the rats were accustomed by being placed on the rod rotating at low speed for at least 60 s. During the test, the rod (6 cm in diameter) was accelerated from 4 to 40 rpm in 300 s and the latency to fall from the rod was measured with a cutoff time of 600 s.

**Brain slice preparation.** Acute brain slices were obtained following halothane anaesthesia and decapitation. The brain was quickly removed and 250–300 µm thick coronal slices containing the striatum or horizontal slices containing the midbrain were cut with a Leica vibratome (VT1200S) using chilled bubbled (95% O₂, 5% CO₂) 'sucrose-based' artificial CSF (aCSF) solution containing (in mM): KCl₃, NaH₂PO₄ 1.25, NaHCO₃ 26, MgSO₄ 10, CaCl₂ 0.5, glucose 25, sucrose 185; ~300 mOsm, pH 7.4). Slices were used after a minimum 40 min recovery period in normal aCSF solution containing (in mM): NaCl 126, KCl 2.5, NaH₂PO₄ 1.2, NaHCO₃ 24, MgCl₂ 1.3, CaCl₂ 2.4, glucose 10 (~290 mOsm, pH 7.4) at 32 ℃.

**Constant potential amperometry.** Amperometric detection of electrically evoked DA release was performed in acute brain slices containing the dorsal striatum. Briefly, the DA-recording carbon fibre electrode (diameter 30 µm, length 100 µm, World Precision Instruments) was positioned near a bipolar Ni/Cr stimulating electrode, to a depth of 50–150 µm into the coronal slice. The imposed voltage (MicroC potentiostat, World Precision Instruments) between the carbon fibre electrode and the Ag/AgCl pellet was 0.55 V. For stimulation, a single rectangular electrical pulse was applied using a DS3 Stimulator (Digitimer) every 5 min along a range of stimulation intensities (20–1000 µA, 20–40 µs duration). In response to a protocol of increasing stimulation[74], a plateau of DA release was reached at maximal stimulation (1000 µA, 40 µs). Signals were digitized with Digidata 1440 A coupled to a computer running pClamp 10 (Molecular Devices). Electrode

calibration was performed at the end of each experiment by bath-perfused DA (0.3–10 μM).

For testing the acute effects of IFN-γ, striatal slices were incubated with recombinant mouse IFN-γ (100–200 ng ml$^{-1}$; R&D Systems) or aCSF for 3 h before amperometric recordings.

The evoked release of DA by amphetamine was evaluated following bath perfusion of 30 μM amphetamine dissolved in aCSF.

**Electrophysiology and Ca$^{2+}$ microfluorometry.** A single horizontal midbrain slice containing the SNpc was transferred in a recording chamber (volume ~0.6 ml) on the stage of an upright microscope (Axioscop 2FS; Carl Zeiss, Germany) and perfused with aCSF (2.5–4.0 ml min$^{-1}$, 32 °C). Whole-cell patch-clamp and conventional single-unit extracellular recordings were conducted on DA neurons in the SNpc[75]. Briefly, for single-unit extracellular recordings, to evaluate the DA neuron spontaneous firing activity, the SNpc DA neurons were identified using the following criteria: (a) location along the ventral border of the medial lemniscus and laterally to the medial terminal nucleus of the accessory optic tract; (b) slow spontaneous firing; (c) long spike duration; d) transient inhibitory response to bath application of DA (30 μM, 2 min). Recordings were performed by slowly moving an aCSF-filled recording electrode in the slice until firing was detected. Electrodes were pulled from thin-wall filamented glass (TW150F4; World Precision Instruments). Spikes were recorded using the $I = 0$ mode of a MultiClamp 700B amplifier with high-pass (0.5 Hz) and low-pass filtering (1 kHz), digitized at 20 kHz sampling rate with Digidata 1322 A and computer using pClamp9 (Molecular Devices). The instantaneous firing frequency from extracellular recordings was obtained using the threshold method (pClamp9). The coefficient of variation was calculated as the ratio of standard deviation/mean inter-spike interval (ISI), for each recording. The sensitivity to DA in extracellular recordings was calculated as percentage of mean instantaneous firing frequency reduction compared to the control firing (before the start of DA application).

Patch-clamp recordings were performed with thin-wall pipettes (4–6 MΩ) filled with a solution containing (in mM): 120 K-gluconate, 20 KCl, 10 HEPES, 2 MgCl$_2$, 4 ATP-Mg$_2$, 0.3 GTP-Na$_3$, 0.2 EGTA (pH 7.2, ~280 mOsm). Whole-cell currents (−60 mV, with MultiClamp 700B) were filtered at 3–4 kHz using the amplifier's in-built low-pass filter, digitized with Digidata 1322 A and computer-saved at a sampling rate of four times the filter frequency. Upon membrane rupture, the cell's Cm was taken online from the membrane seal test function of pClamp 9 (−5 mV step, 15 ms). $I_h$ currents were induced by hyperpolarizing voltage steps (from −60 to −120 mV at −20 mV intervals, 1 s duration). After-hyperpolarization currents (AHC) were recorded using a single depolarizing voltage step (from −60 to 0 mV, 100 ms) and measuring the area under the outward current. The activation (for $I_h$) and deactivation kinetics (for AHC) were analysed with exponential fits. Currents mediated by DA (1, 10, and 30 μM, 2 min bath application) or Baclofen (1 and 10 μM, 3 min) were analysed for peak amplitude.

In current-clamp mode, stepped current injections (600 ms, 50 pA increments, from 200 to −200 pA) were used for obtaining action potential (AP) numbers at supra-threshold responses and current/voltage plots at sub-threshold responses. The threshold potential was taken from the maximum of the second derivative of membrane potential by time, corresponding to the inflection point at the start of the AP. Membrane resistance (Rm) was calculated from the slope after linear regression of current/voltage curves. Sag ratio was the ratio of the steady-state versus peak potential during sub-threshold responses to −200 pA current injections. Current clamp recordings of current-induced spiking were obtained from a resting membrane potential kept to −50 mV by current injection.

During all experiments, the membrane access resistance was repeatedly monitored and recordings in which it exceeded over 25% were discarded. No liquid junction potential correction was applied.

Measurements of intracellular free Ca$^{2+}$ concentration were performed in whole cell using pipettes (2–5 MΩ) filled with (mM): 145 K-gluconate, 0.1 CaCl$_2$, 2 MgCl$_2$, 10 HEPES, 0.75 mM EGTA, 0.25 Fura-2 K$^+$ salt (ab142777, Abcam), 2 ATP-Mg2$^+$ and 0.3 GTP-Na$^+$ (pH 7.3, 280 mOsm). Cells were illuminated using a monochromator-based system (Till Photonics), which provided 340 and 380 nm excitation wave-lengths. Fluorescence ratios (R) were calculated by the specific fluorescence values F$_{340}$ and F$_{380}$ emitted by ROI and background (BK) at 340 and 380 nm excitation wavelengths with the relationship:

$$R = (F_{340}ROI - F_{340}BK)/(F_{380}ROI - F_{380}BK) \qquad (1)$$

and were then converted to intracellular free Ca$^{2+}$ concentration using the relationship:

$$[Ca^{2+}] = K_D \times \beta \times (R - R_{min})/(R_{max} - R) \qquad (2)$$

where $K_D$ is the effective Fura-2 dissociation constant (225 nM), $\beta$ is the ratio of 380 nm excitation florescence at zero and saturating Ca$^{2+}$ levels, $R_{min}$ and $R_{max}$ values were obtained in situ by exposing perforated cells (1 μM ionomycin) to Ca$^{2+}$-free (0 mM Ca$^{2+}$, 1 mM EGTA) and 3 mM Ca$^{2+}$-containing bath solutions.

For experiments examining the basis of Ca$^{2+}$ increase in Syn DA neurons, cells were first bathed with normal aCSF to obtain the control levels of Ca$^{2+}$ at −60 mV and were then exposed to either Ca$^{2+}$-free aCSF (to examine contribution of VGCCs) or to 10 μM CPA (to examine ER stores) for at least 2 min. Experiments

with isradipine were performed in separate slices that were incubated for 1 h with 200 nM isradipine, to allow its diffusion within the lipid bilayer[52].

**Immunohistochemistry and immunofluorescence.** Rats were deeply anesthetized by i.p. injections of xylazine (Rompun; 20 mg ml$^{-1}$, 0.5 ml kg$^{-1}$ Bayer) and tiletamine/zolazepam (Zoletil; 100 mg ml$^{-1}$, 0.5 ml kg$^{-1}$; Virbac) and perfused transcardially with 4% paraformaldehyde in phosphate buffer (PB; 0.1 M, pH 7.4). The brains were removed and postfixed in paraformaldehyde at 4 °C and then immersed in 30% sucrose solution at 4 °C until sinking. The brains were then cut into 30-μm-thick coronal sections using a freezing microtome, to be used for immunohistochemistry and immunofluorescence[76,77].

Brain sections for immunohistochemistry were incubated at 4 °C with the primary anti-TH antibody diluted in PB containing 0.3% Triton X-100. After washing, sections were incubated with a biotinylated secondary antibody (Jackson Immunoresearch Laboratories) followed by the avidin–biotin–peroxidase method (Vectastain, ABC kit; Vector) and using chromogen 3,30-diaminobenzidine (Sigma). Sections were counterstained with Nissl, dehydrated and coverslipped with Entellan (Sigma). These sections were subsequently used for quantification of TH$^+$ and TH$^-$ neurons in the midbrain (see stereological cell count below).

For immunofluorescence (TH/GFAP, TH/Iba1, DAPI/GFAP, DAPI/Iba1), brain sections were incubated overnight with primary antibodies in PB containing 0.3% Triton X-100 and then incubated for 2 h at room temperature with secondary antibodies. For 3D reconstruction of Iba1 or GFAP cell, images were taken as Z-stacks and these Z-stack images were then processed by maximum intensity projection. All samples were acquired with the same laser settings. TH levels in the dorsolateral striatum were quantified with ImageJ (http://imagej.nih.gov/ij/) as mean fluorescence intensity (F) on a defined area (A).

Primary antibodies: TH (1:700, Millipore; MAB318; RRID: AB_2201528), GFAP (1:200, DAKO, Z0334; RRID: AB_2314535), Iba1 (1:400, Wako #019-19741; RRID: AB_839504). Secondary antibodies: Alexa Fluor 488 donkey anti-mouse IgG (1:200; Thermo Fisher Scientific Cat# R37114, RRID: AB_2556542) and Alexa Fluor 555 donkey anti-rabbit IgG (1:200; Thermo Fisher Scientific Cat# A-31572, RRID: AB_162543). The sections were counterstained with DAPI and examined under a confocal laser-scanning microscope (LSM700, Zeiss). The specificity of the immunofluorescence labelling was confirmed by the omission of primary antibodies and the use of normal serum instead (negative controls).

**Sholl analysis.** We analysed Iba1$^+$ cells (microglia) and GFAP$^+$ cells (astrocytes) in the SNpc, striatum, dorsal hippocampus and pontine nuclei[49]. Cells were imaged with an optical microscope (DMLB; Leica) equipped with a motorized stage and a camera connected to Neurolucida 7.5 software (MicroBright-Field) that allowed for quantitative 3D analysis of the entire cell compartment. Only cells that showed intact processes unobscured by background labelling or other cells were included in cell reconstructions. We evaluated the cell body area and perimeter, number of intersections, number of nodes (branch points) and endings and total length of processes. To account for changes in the cell's complexity in relation to distance from the soma, concentric circles (radii) were spaced 10 μm apart, originating from the soma and the number of branch points and endings, processes that intersected the radii and process length were measured as a function of the distance from the cell soma for each radius. Overall, fifteen cells per animal were selected randomly for analysis, and all data were subsequently averaged for each rat. Subsequently all animals were averaged per experimental group (for the plots showing relationship to radial distance from soma) or shown as individual points (for plots of soma area and perimeter).

**Dot blot.** Animals were anaesthetized, decapitated and the brains regions of interest were dissected and collected. Thus, sequential extraction of α-syn was performed. Each sample was weighed and homogenized in freshly prepared, ice-cold TBS consisting of 20 mM Tris-HCl, 150 mM NaCl, pH 7.4 and protease inhibitor cocktail at 5:1 (TBS volume/brain wet weight) and homogenized. The homogenate was spun at 175,000 × g in a TLA100.2 rotor on a Beckman TL 100 centrifuge. The supernatant (TBS extract) was stored at −80 °C.

For dot blot quantification of α-syn aggregates, 1 μg of tissue homogenate from the specified regions and fractions was spotted in 1 μl volume onto 0.45 μm nitrocellulose membranes. Immunoblotting analysis was performed using a chemiluminescence detection kit. The relative levels of immunoreactivity were determined by densitometry using the ImageJ software. Primary antibodies: anti-Alpha-synuclein filament antibody [MJFR-14-6-4-2] (1:1,000, Abcam, ab209538; RRID:AB_2714215); anti-Actin (1:60,000, Sigma-Aldrich, A5060; RRID: AB_476738). For a full blot, see also the Sounce data file.

**Stereological cell count.** Stereological cell counting was performed for unbiased estimates of total number of TH$^+$ and TH$^-$ neurons in the SNpc and VTA[78], as well as for numbers of Iba1$^+$ cells (microglia) and GFAP$^+$ cells (astrocytes) in the SNpc, striatum, dorsal hippocampus and pontine nuclei. Midbrain sections processed for TH immunohistochemistry were used for bilateral cell counting or TH$^-$ and TH$^+$ neurons in the SNpc and VTA. The boundaries of these areas in the rat brain were defined by TH staining and area distinction was performed according to a rat brain atlas. Cell counting for GFAP$^+$ or Iba1$^+$ cells was done on slices

processed by immunofluorescence (TH/GFAP, TH/Iba1, DAPI/GFAP, DAPI/Iba1; see above).

Cell counting was performed by an optical fractionator stereological design using the Stereo Investigator System (MicroBrightField Europe e.K.). A stack of MAC 5000 controller modules (Ludl Electronic Products, Ltd) was interfaced with an Olympus BX50 microscope with a motorized stage and a HV-C20 Hitachi digital camera with a Pentium II PC workstation. A 3D optical fractionator counting probe (x, y, z dimension of $50 \times 50 \times 25$ μm) was applied. The brain areas of VTA, SNpc, dorsolateral striatum, dorsal hippocampus or pontine nuclei were outlined with a 5x objective and neurons were marked with a ×100 oil-immersion objective. Neurons were considered positive for the cell marker if they showed cytoplasmatic immunoreactivity. The total cell number for each brain area was estimated according to the formula:

$$N = SQ \times \frac{1}{ssf} \times \frac{1}{asf} \times \frac{1}{tsf} \qquad (3)$$

where SQ is the number of neurons counted in all optically sampled fields of the area of interest, ssf is the section sampling fraction, asf is the area sampling fraction and tsf is the thickness sampling fraction.

**Collection of rat CSF and plasma and flow cytometry**. CSF (about 0.5 ml) and blood (about 0.5 ml) sampling from rats was performed during deep anaesthesia (200 mg kg$^{-1}$ Rompun and 800 mg kg$^{-1}$ Zoletil, i.p.) while the animal was positioned in a stereotaxic apparatus. A 23 G needle was inserted into the cisterna magna for CSF collection without making any incision at this region. The non-contaminated sample was drawn into the syringe by simple aspiration. Arterial blood was sampled from the heart of the same animal. CSF and plasma samples were stored at −80 °C until use.

For RvD1- or saline-treated rats, collection of CSF and plasma was performed 24 h after the last injection. For analysis of peripheral blood, cells were lysed with 1× red blood cell lysis buffer for 10 min at room temperature and then stained for 30 min at 4 °C with fluorochrome-conjugated antibodies: anti-granulocytes-FITC (1:100, REA535, Miltenyi Biotec Cat# 130-108-119; RRID: AB_2651885), CD3-PE-Vio770 (1:100, REA223, Miltenyi Biotec Cat# 130-103-772; RRID: AB_2657102), CD45RA-PE (1:100, OX-33, Biolegend Cat# 202307; RRID: AB_314010), CD11b/c-APC (1:100, clone REA325, Miltenyi Biotec Cat# 130-120-214; RRID: AB_2752041), MHC-II-VioBlue (1:100, REA510, Miltenyi Biotec Cat# 130-107-818; RRID: AB_2652891), CD68-APC-Vio770 (1:100, REA237, Miltenyi Biotec Cat# 130-103-366; RRID: AB_2659019). All samples were re-suspended in running buffer and 50,000 events were acquired on the Cytoflex (Beckman Coulter) flow cytometer. Immune cell populations were gated as follows: granulocytes (anti-granulocytes+), T cells (CD3+/CD45RA−), B cells (CD45RA+/CD3−) and monocytes (CD11b/c+). The percentage of expression of MHC-II and CD68 was then evaluated on CD11b/c+ monocytes. Flow cytometry analysis was performed using FMO for compensation for all fluorochromes and also with the respective isotypes of the antibodies used and using Flowjo V8 software. Data are expressed as % of cells positive for the given fluorochrome-conjugated antibodies within total cells or within CD11b negative or CD11b positive cells.

**Collection of human CSF and plasma**. For collection of human CSF and plasma, 16 subjects (8 PD patients and 8 age-matched healthy controls) were consecutively recruited at the Neurology Department of the Tor Vergata University Hospital (Rome, Italy). PD was diagnosed according to the British Parkinson's Disease Society Brain Bank (UK-PDSBB) criteria. All PD patients were at early disease stage and were also untreated (de novo, not taking levodopa, monoamine oxidase inhibitors or DA receptor agonists). The control group included age-matched subjects without degenerative and inflammatory diseases, not presenting motor or cognitive disturbances (e.g. patients with psychogenic disorders). Exclusion criteria for this study were: age younger than 50 or older than 80, dementia (Mini-Mental State Examination, MMSE score < 24), treatment with anti-inflammatory drugs in the last month, history of autoimmune/inflammatory diseases, cancer, thyroid disorders, diabetes or any other acute condition. Ethical approval and guidelines for the study protocol were obtained by the Tor Vergata University Hospital. All enroled subjects, after signing an informed consent, underwent a diagnostic and experimental study protocol including laboratory tests, full neurological examination, standard neuropsychological evaluation by MMSE, brain magnetic resonance imaging and lumbar puncture for CSF analysis. PD patients were further evaluated with the Unified Parkinson's Disease Rating Scale (UPDRS) part III and Hoehn and Yahr (H&Y) scale for motor signs.

Lumbar puncture was performed in the morning following standard procedures and CSF (6–8 ml) samples was taken in polypropylene tubes without preservatives, gently mixed, and immediately carried in ice to the central lab. The CSF samples used for analysis were centrifuged at 2,000 rpm at 4 °C for 10 min, aliquoted in polypropylene vials and stored at −80 °C until use. CSF samples containing >500 erythrocytes μl$^{-1}$ were excluded. Blood samples were taken at the same time, to evaluate the CSF/blood albumin ratio and blood–brain barrier integrity. Plasma samples were collected in BD Vacutainer tubes (Becton Dickinson) using EDTA as anti-aggregant, centrifuged at 3500 rpm for 5 min, transferred in vials and stored at −80 °C until use. Levels of neurodegeneration biomarkers (Table 1), in particular β-amyloid 1-42 and β-amyloid 1-40, total and phosphorylated tau were measured

in the CSF using commercially available kits (INNOTEST hTau, INNOTEST phospho-tau for 181p, INNOTEST β-amyloid 1-40, Lumipulse G β-amyloid 1-42).

**Detection of cytokines**. The levels of cytokines in CSF and plasma from rats and human subjects were measured by standard sandwich ELISA through custom-made magnetic Luminex multiple assays (R&D Systems), according to the manufacturer's instructions and read on a Luminex200 (Life Technologies).

**Detection of resolvins**. The levels of RvD1 and RvD2 in CSF and plasma from rats and human subjects were measured with quantitative competitive ELISA kits and validated in Cayman's EIA Buffer, based on the competition between free RvD1 Tracer and RvD1-specific rabbit antiserum binding sites[49]. The amount of RvD1 or RvD2 Tracers that were able to bind to the rabbit antiserums was inversely proportional to the concentration of free RvD1 or RvD2 in the wells. The detection of the rabbit antiserum-RvD1 or RvD2 was based on a modified sandwich ELISA and the absorbance read between 405 and 420 nm of a VarioScan FLASH (Thermo Scientific; assay sensitivity 15 pg ml$^{-1}$).

**Sample size, randomization and blinding**. The number of samples in each group and for each experiment was determined based on published studies. All randomization was performed by assigning a random number to each animal and using a random number table. All data were collected by researchers that were blind to the genotype or pharmacological treatment of each animal.

**Statistical analysis**. All statistical analysis was performed with GraphPad Prism (v7.00). Data comparing the two different age groups (behaviour, amperometry, IFN-γ, RvD1 and RvD2 levels) were analysed by ordinary two-way analysis of variances (ANOVA) with genotype (WT versus Syn) and age (2 versus 4 months) as independent factors. Ordinary two-way ANOVA was also used for: TH$^+$ and TH$^-$ cell numbers in the SNpc and VTA (analysed for cell-type and genotype); amperometry data with IFN-γ (analysed for genotype and slice treatment); dose response curves to DA and Baclofen (analysed for genotype and drug concentration); all other data following sub-chronic RvD1 treatment (analysed for genotype and treatment).

Two-way repeated measures ANOVA was used for: neuron excitability data (AP plots, analysed for genotype and drive current) and current–voltage plots (analysed for genotype and drive current); cell parameters obtained from Sholl analysis (analysed for genotype and radial distance from soma for naïve animals, or for treatment and radial distance for treated rats).

All post hoc comparisons following ANOVAs were assessed with Bonferroni's test.

The rest of the data (WT versus Syn rats or control versus PD patients) were checked for normality using the Shapiro-Wilk and D'Agostino & Pearson normality tests and analysed accordingly with two-tailed parametric or non-parametric tests (Welch's t-test or Mann–Whitney test, respectively). See figure legends for more details. Values of $P \leq 0.05$ were considered to be statistically significant (shown in Figures as *$P \leq 0.05$, **$P \leq 0.01$ and ***$P \leq 0.001$). In figures, in box-and-whisker plots the centre lines denote median values, edges are upper and lower quartiles, whiskers show minimum and maximum values and points are individual experiments. All other data are presented as mean ± s.e.m.

**Reporting summary**. Further information on research design is available in the Nature Research Reporting Summary linked to this article.

## Data availability
The datasets generated and analysed during the current study are available in the Source Data file accompanying the paper; raw data can be obtained from the corresponding author on request.

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

## Acknowledgements

N.B.M. was supported by the Italian Ministry of Health (Research Grant: RF-2018-12365509) and by the ONLUS Foundations 'Fondazione Baroni' and 'Fondazione il Fulcro'. M.D.A. was supported by grants from the Italian Ministry of Health (Young Investigator's Award: GR-2011-02351457; Research Grant: RF-2018-12365527) and from the Alzheimer's Association (Grant: AARG-18-566270). V.C. was supported by the Italian Multiple Sclerosis Foundation (Grant: FISM 2017/R/8) and by the Italian Ministry of Health (Young Investigator's Award: GR-2016-02362380). C.N.S. was supported by the National Institutes of Health (Grant: R01GM38765). A.N. and P.K. were supported by Post-doctoral Fellowships by the Collegio Ghislieri and the Veronesi Foundation, respectively. We thank Drs Riviello and Wirz for their assistance with animal caring.

## Author contributions

A.C., P.K., V.C., M.D.A. and N.B.M. conceived and designed the study; L.L.B., A.N. and M.T.V. designed and carried out the immunohistochemistry, immunofluorescence, dot blots, DA neuron counting and Sholl analysis experiments; M.D.A. and M.T.V. supervised these experiments; M.T.V. and A.L. carried out CSF and blood collection from rats; A.C., M.F., P.K. and F.V. designed and performed electrophysiological/amperometric recordings; N.B.M. supervised the electrophysiology experiments; B.P., V.G., F.C., G.A., G.M. and G.N., V.Cal. designed and performed the behavioural experiments; P.C. supervised the behavioural experiments; N.C. and O.R. provided the rats; V.C. and A.L. designed and performed detection of cytokines and resolvins, performed the pharmacokinetics analysis and immunological phenotyping, analysed and interpreted the immunological data; G.D.L., T.S., G.S.C., S.B. and A.P. recruited patients, performed medical examinations and diagnosis, and CSF and blood collection. P.K., A.C., V.C., M.D.A. and N.B.M. wrote the manuscript. C.N.S. provided scientific suggestions and revised the manuscript. All authors discussed results and commented on the manuscript.

## Additional information

**Competing interests:** The authors declare no competing interests.

