## [Peer Review File · Nature Communications]

Reviewers' comments:

Reviewer #1 (Remarks to the Author):

In this paper, Cordella et al., show that a neuroinflammatory mediator RvD1 prevents the onset of Parkinson's symptoms in a transgenic rat model that overexpresses full-length human alpha-synuclein. Overall, it is indeed an intriguing study that links alpha-synuclein to microglia in PD. The study is particularly attractive with a translational component that may show potential for resolvins as novel therapeutics. While the study is interesting and the experimental findings are novel and potentially important, the analyses provided are rather crude. Although the study would benefit from having a more robust longitudinal (e.g. additional age points to be added) analysis to track the disease progression, it would significantly delay the publication of the current work.

Major

- The authors should look into basal Ca²⁺ levels in dendrites of SNpc neurons since somatic and dendritic Ca²⁺ affect cellular excitability in different ways. The authors should discuss or look into the potential reasons for high levels of basal Ca²⁺ in Syn rats. Is this due to voltage-gated calcium channels, internal calcium stores or calcium buffering?
- Additionally, SNpc neurons have somatic and dendritic Ca²⁺ oscillations (Guzman et al., 2010) and hence it is unclear as to what the authors mean by basal levels.
- Would this high Ca²⁺ be the reason why afterhyperpolarization currents in Figure 2 are larger in Syn rats? It would be good to report the actual Ca²⁺ concentrations since calibration was performed by the authors.
- In line 419, the authors state that the high conductance of SK to be at fault. Is it a change in single channel conductance or the expression level of SK channels that lead to differences between WT and Syn rats? Importantly, this was not directly tested. It is more likely that the observed difference is a consequence of an altered Ca²⁺.
- The authors should note that the sample sizes in this study are low in general. For instance, for the comparison of CSF and plasma RvD1 or 2 between WT and Syn rats particularly in the 4-month group. This is concerning as the variability is high in ELISA tests.
- The authors should refrain from using ABC-DAB staining (line 220) for quantification analysis as the approach is highly non-linear and irreproducible. More importantly, it is unclear as to what the chromo-DAB method was used for. The authors mention it was used for immunohistochemistry, while Alexa secondaries were used for all the figures shown in the manuscript.
- The authors should look into the possibility that an increase in microglia numbers could be triggered by dopamine neuronal degeneration.
- It is unclear if only microglia but not astrocytes become reactive in the 4-month-old alpha-synuclein rats. Additionally, the authors should move supplementary figure 2e to the main text since it is important to show the staining for astrocytes early on.
- The authors should address if areas outside the basal ganglia show a similar increase in microglia numbers or change in morphology.
- Similarly, it is unclear if the observed microglial activation is region-specific and if that correlates with the expression of alpha-synuclein.
- The authors state that there is a relationship between alpha-synuclein, microglia, and pro-inflammatory cytokines and suggest that this interaction occurs early on and that mechanisms driving resolution of inflammation may be at fault. However, these potential mechanisms are not discussed.
- To promote the translation value of RvD1 treatment, the authors should thoroughly examine the pharmacokinetics and bioavailability of RvD1 in rats/humans. It is unclear how the treatment concentration was chosen. To provide rigor, it is important to show how neuroinflammatory signaling cascades are perturbed/restored with different experimental conditions (i.e., WT vs Syn +/- RvD1).
- The patient cohort included in this study ranges from 50 to 80 years old. It is unclear how that compares to the 4-month-old rats. Importantly, more clinical information, such as the UPDRS rating of patients should be reported.

Minor

- The authors state in the title that RvD1 prevents neuroinflammation and thereby onset of PD. However, they state in the discussion (line 554) that it delays the onset of PD. Based on the data shown, 'prevents' would seem more appropriate. Without looking at the disease progression more systematically/thoroughly, the word "delays" is confusing.
- Line 42-43 is a confusing sentence (family of omega 3 fatty acids-derived specialized pro-resolving mediators that include resolvins).
- The reconstructed cells should have the soma filled with a darker shade so as to help the readers.
- The authors should not assume that all data have a normal distribution and should determine if non-parametric tests are needed.
- The authors should state if the Syn rats are from Sprague-Dawley background.
- The authors should explain why the kinetics of baclofen elicited currents are different from that of DA-induced currents.

Reviewer #2 (Remarks to the Author):

This study investigates the role for resolving D1 (RvD1) in a transgenic rat model of Parkinson disease and provides compelling evidence that RvD1 is involved in the disease activity. The authors provide detailed characterization of the relatively new rat model based on transgenic expression of alpha-synuclein, including data on behavior (motor and non-motor), neuron loss, and dopaminergic neuron function. In this model, they find evidence for microglial activation and altered levels of RvD1 in both plasma and CSF, and report that both the physiological abnormalities and the morphological activation of microglia are rescued by administration of RvD1. This work is strengthened by the inclusion of human data, which demonstrates that RvD1 is decreased in the CSF and plasma of early Parkinson patients. Overall, this article provides evidence that resolvins are important in the pathogenesis of Parkinson disease, a novel idea worthy of further study.

There are some issues that should be addressed:

- 1) The rat model, while thoroughly characterized at the behavioral and electrophysiological levels, does not show neurodegeneration at early time points. Authors cite a paper that states neurodegeneration occurs later in the model, but the citation is incorrect and references a review article by Ransohoff et al.
- 2) The authors do not specify how the dose of RvD1 was determined. Authors should provide a clear rationale and pharmacological data for dosing, or cite a relevant paper if the dosing has already been established in the literature.
- 3) In Figures 6 and 7, the authors administer RvD1 to correct the various phenotypes found in the rat model. However, it would be important to determine whether this treatment in fact corrects the deficiencies of RvD1 in CSF, plasma and brain.
- 4) Finally, the authors use bar graphs that display each data point within the bar. However, the number of data points do not always match the reported number of animals used. It is possible that multiple animals fall at the same point on the graph-authors should use a graph format that displays similar data points side by side.
- 5) It seems that the authors have mixed up which figures they are referring to when discussing Figures 5 c, d in the text. Also, Figure 5e is never mentioned in the text.

Reviewer #3 (Remarks to the Author):

In the manuscript Cordella et al. approach early changes in a Parkinson's disease (PD) model based on human α -synuclein (α -syn) expression in rats, and the potential of resolvin D1 (RvD1) as a neuroprotective drug that can resolve neuroinflammatory changes associated to α -syn and protect neurons.

The BAC human α -syn rats, express the protein under the human promoter. These rats have been previously shown to exhibit certain dopaminergic deficits and α -syn pathology, although these changes were observed at later age. In the manuscript, the earliest signs of change are observed at 4 months, so authors selected 2 and 4 months as time points for the study.

The authors describe early (4 months) changes in behavior and in the firing pattern of dopaminergic (DAergic) nigral cells that were associated to differences in DA release in striatum and changes in DA sensitivity possible related to GIRK/Kir3 channels. This was observed in the absence of any cellular (in substantia nigra (SN)) or axonal (in striatum) DAergic loss.

The authors relate these changes to early microglia response in SN and striatum, as shown by increase in the number of microglia and in their morphological complexity. In addition, they report increased IFN γ in CSF. Interestingly, these inflammatory changes were associated to elevated RvD1 in CSF, while the plasma level of RvD1 was decreased, suggesting an opposing event occurring in periphery and in brain at early stages. However at 18 months of age, the RvD1 was decreased both in CSF and plasma. Interestingly a similar decrease in RvD1 is also found in CSF and plasma from PD patients.

The team attempts to examine if treatment with RvD1 can result in neuroprotection. After 2 months of treatment all reported changes were abolished and the animals appeared similar to WT, suggesting a therapeutic ability for the RvD1 in α -syn induced neurodegeneration and inflammation.

While the early neurodegenerative signs described in the model and its correlation with immune response is relevant, and the protection with RvD1 very promising, several points need to be addressed in the work:

1. Although the quantification of microglia morphological profile is adequate and nicely presented. The excessively simplistic approach for the quantification of number of microglia (and astroglia) in SN and in striatum is very poor. The authors should approach a stereological quantification that more accurately can give a realistic estimation of total number of microglia through the two brain areas: SN and striatum.

2. In addition, the analysis of other inflammatory markers should have brought some light to the study. Such as MHCII, CD68, TLR2 or TLR4 expression, monocyte and lymphocytes infiltration in striatum or SN.

3. Also are these changes observed found only for dopaminergic nigro-striatal areas or where other areas in brain also showing signs of inflammation?

4. The authors observed change in RvD1 in peripheral as well as in brain. The literature suggest changes in the immune system in PD includes also peripheral immune changes. However the authors do not address whether the observed changes are due to peripheral immune events (except for the selected panel in plasma) that might ultimately reflect in brain changes.

In general I think the study requires a better characterization of immune response, in brain and in periphery. This seems especially relevant based on the opposite changes observed in RvD1 at early ages with increase in CSF while decrease in plasma.

5. How were the RsD1 levels in brain and plasma after the 2-month RsD1 treatment?

6. Analysis of pathological a-syn should also be included with markers such as Phospho-a-syn and antibodies such as MJF14 (please see the MJ Fox Foundation resources web-page) at both ages and treatments. Are these changes associated to detectable aggregated or pathological a-syn or were they prior to any detectable pathology?

7. The reference list includes 109 references although in the text only 75 are cited. In fact the citation given in text never corresponded to the article with such number in the reference list. This made the reading process extremely complicated.

8. The discussion regarding the putative role of resolving in the model and the protection could be improved (although due to the chaotic reference list, it was difficult to follow). The authors should reflect or suggest putative mechanism. Is this effect of RvD1 a consequence of its effect in neurons or in microglia? The reference regarding the RvD1 protection in the MPP+ model (in text indicated as #74 and in reference list #107) suggest an effect on neurons directly. However, the RsD2 in the LPS model seems to exert the protection through the microglia (reference in text indicated as #75 and in reference list #106)

RsD have been related to inhibition of a TLR signaling through glycogen synthase kinase 3 (GSK3) (PMID:26878867 PMID:16007092) and also a-syn activation of microglia seems related to TLR4 (PMID:23108585) and TLR2; and furthermore TLR2 seems also relevant in neurons in PD (PMID:27888296).

Resolvins have been also associated to T-cell activation (PMID:27559094), and T-cells have been related to PD and a-syn (PMID:26018603). In that regard, IFN-g is an important mediator of T-cell response.

OTHER

1. In page 10 the authors indicate "15 cells per animals and per experimental group were selected..... "

However, little is told about how the data is handled later. Were the markers studied averaged per animal and subsequently per group?

Based on fig 4 only 3 animals were used. Despite the differences found this is very small number, in fact this is true for many of the measurements which has 4 or less animals used.

2. In graphs with individual numbers represented by points, they should be drawn in a way that two points of equal value can be distinguished (i.e distributed not only vertically but also horizontally).

In that regard, please in fig 8 indicate out of the total n number how many patients showed detectable levels of each marker, since in the graphs this is not possible to really asses.

3. In figure 6F the authors show that CSF IFNg in wt saline is significantly different than a-syn saline, but based on the graph that seems unlikely. However, the graph suggest that the treatment leads to decrease of IFNg in the WT animals.

4. Page 20 In the text describing the changes of RvD1 and RvD2 with age, the authors should indicate : compare Fig5b vs. 5d and 5c vs.5e, to help the reader.

5. It is unclear to this reviewer whether the statistical approach used is strictly correct or not. The authors indicate that they approach each factor as independent variable, irrespective that the factors show interaction, which does not seem correct. A statistic expert should comment in that particular point.

We would like to thank the Reviewers for their constructive comments, as we believe that the resulting changes to the manuscript helped to improve the quality of our work and strengthened our results. We also apologize for the mess with the references, these are now corrected in the revised version. In line with the journal's policy, we provide a checklist regarding statistical analysis, and a 'Source Data file' with all the raw data. Additionally, the Abstract and main text have been changed to comply with the journal's format requirements.

Below is a point-to-point reply to all the comments raised by the Reviewers; In the main text all the relevant changes are now highlighted for easiness of reading.

POINT-BY-POINT REPLY TO REVIEWER #1

Reviewer #1:

In this paper, Cordella et al., show that a neuroinflammatory mediator RvD1 prevents the onset of Parkinson's symptoms in a transgenic rat model that overexpresses full-length human alpha-synuclein. Overall, it is indeed an intriguing study that links alpha-synuclein to microglia in PD. The study is particularly attractive with a translational component that may show potential for resolvins as novel therapeutics. While the study is interesting and the experimental findings are novel and potentially important, the analyses provided are rather crude. Although the study would benefit from having a more robust longitudinal (e.g. additional age points to be added) analysis to track the disease progression, it would significantly delay the publication of the current work.

Point 1 raised by Reviewer #1:

-The authors should look into basal Ca²⁺ levels in dendrites of SNpc neurons since somatic and dendritic Ca²⁺ affect cellular excitability in different ways. The authors should discuss or look into the potential reasons for high levels of basal Ca²⁺ in Syn rats. Is this due to voltage-gated calcium channels, internal calcium stores or calcium buffering?

Reply:

-We thank the Reviewer for the recommendations as the new experiments greatly improve the quality of our work. Although we lack the facility for performing two-photon dendritic Ca²⁺ imaging, following the Reviewer's recommendations we performed a series of new conventional Ca²⁺ imaging experiments to investigate the source of high levels of cytoplasmic Ca²⁺ in Syn rats. These experiments include: (i) investigating the contribution of voltage-gated calcium channels to cytosolic Ca²⁺ signals, using 0 extracellular Ca²⁺ or incubation of slices with the L-type channel

blocker isradipine (1h incubation, 200 nM as in Guzman et al., 2009); (ii) investigating the contribution of Ca²⁺ release from intracellular stores following induction by CPA in brain slices from 4 month-old WT and Syn rats.

The results of these new experiments are shown in **Fig. 3 (new panels e-g)** and the Methods section is expanded to include the new experiments. Overall, it appears that the contribution of VGCCs in the increased somatic [Ca²⁺] in Syn rats at resting conditions (-60 mV) is minimal since both application of the Ca-free ACSF or incubation of slices with isradipine failed to cancel the difference between WT and Syn animals. This is in line with the fact that at -60 mV there is a shut-down of L-type channels (Philippart et al., J Neurosci. 2016;36(27):7234-45). Of course, this does not exclude that Ca²⁺ entry from dendritic VGCCs during pacemaking could contribute to the accumulation of Ca²⁺ and to dopaminergic dysfunction in Syn rats (works by the Surmeier group; Mosharov et al., Neuron. 2009;62(2):218-29), but the low resolution of our technique does not permit to study this further (this is also now discussed in the revised text).

On the other hand, the induced release of Ca²⁺ from stores by CPA was much lower in Syn rats, suggesting a reduced capacity of stores to absorb Ca²⁺ in Syn rats, which can explain the increased cytoplasmic content; this is further discussed in the text.

During these experiments we've also confirmed the difference in cytoplasmic Ca²⁺ levels between genotypes; the new data are pooled together with the old ones in **Fig. 3d**.

Point 2 raised by Reviewer #1:

-Additionally, SNpc neurons have somatic and dendritic Ca²⁺ oscillations (Guzman et al., 2010) and hence it is unclear as to what the authors mean by basal levels.

Reply:

The Referee is correct; by basal levels we mean somatic Ca²⁺ levels during voltage-clamp at -60 mV. This is now changed in the text.

Points 3-4 raised by Reviewer #1:

-Would this high Ca²⁺ be the reason why afterhyperpolarization currents in Figure 2 are larger in Syn rats? It would be good to report the actual Ca²⁺ concentrations since calibration was performed by the authors.

-In line 419, the authors state that the high conductance of SK to be at fault. Is it a change in single channel conductance or the expression level of SK channels that lead to differences between WT and Syn rats? Importantly, this was not directly tested. It is more likely that the observed difference is a consequence of an altered Ca²⁺.

Reply:

Indeed, the Referee is correct, many papers argue for a link between increased cytoplasmic [Ca²⁺] and changes in the apamin-sensitive afterhyperpolarization (Fiorillo & Williams, Nature 394, 1998; Seutin et al., J. Neurophysiol. 83, 2000; Xia et al., Nature 395, 1998; Yoshizaki et al., J. Physiol. 486, 1995). We've now modified the text regarding afterhyperpolarization and SK channels to argue that the observed changes in SK-mediated currents in Syn rats are likely to be secondary to increased cytoplasmic [Ca²⁺] (see pg.8 and Discussion). Additionally, all plots previously showing Fura2 ratios are now converted in cytoplasmic [Ca²⁺].

Point 5 raised by Reviewer #1:

-The authors should note that the sample sizes in this study are low in general. For instance, for the comparison of CSF and plasma RvD1 or 2 between WT and Syn rats particularly in the 4-month group. This is concerning as the variability is high in ELISA tests.

Reply:

In line with the Reviewer's suggestion, to increase the sample sizes we performed new measurements of CSF and plasma levels of INF- γ , RvD1 and RvD2 in 2- and 4-month old WT and Syn rats. These results are pooled together with the old ones and reported in **new Fig. 4g, 5c, 5d**. Additionally, we performed new treatments of rats with RvD1 to increase the sample size of CSF levels of INF-gamma (**new Fig. 6f**) and to measure levels of RvD1 in CSF and plasma following treatment (**Supplementary Fig. 5c,d**). Similarly, in line with the request of Reviewer 3 (Minor point 1), we've also increased the sample sizes for the analysis of Iba1+ and GFAP+ cells in the SNpc and striatum (**new Fig. 4, Supplementary Fig. 3 and 4**).

Point 6 raised by Reviewer #1:

- The authors should refrain from using ABC-DAB staining (line 220) for quantification analysis as the approach is highly non-linear and irreproducible. More importantly, it is unclear as to what the chromo-DAB method was used for. The authors mention it was used for immunohistochemistry, while Alexa secondaries were used for all the figures shown in the manuscript.

Reply:

We apologise for the confusion. The ABC kit was used only for immunohistochemistry, to quantify the *number* of TH+ and TH- neurons in the SNpc and VTA of WT and Syn rats (**Fig. 1f**; in this concept, even very weak TH staining is sufficient to mark the neuron as TH+ for cell counting) and not for quantifying the *level* of TH protein (which would indeed be affected by the efficacy of the staining).

The Alexa secondaries were used for immunofluorescence experiments (all other relevant Figures). The Methods section is now corrected.

Point 7 raised by Reviewer #1:

-The authors should look into the possibility that an increase in microglia numbers could be triggered by dopamine neuronal degeneration.

Reply:

Indeed, we had considered this possibility early on, and for this reason we had performed the stereological quantification of TH⁺ (and TH⁻) neurons in the midbrain and the quantification of TH protein levels in the striatum (experiments shown in **Fig. 1**), as an indication of dopamine neuron degeneration. However, at 4 months of age the degeneration is still absent from the SNpc (and VTA), and TH levels in the striatum are normal, whereas microglia numbers are already altered, suggesting that the change in microglia is not triggered by degeneration.

Point 8 raised by Reviewer #1:

- It is unclear if only microglia but not astrocytes become reactive in the 4-month-old alpha-synuclein rats. Additionally, the authors should move supplementary figure 2e to the main text since it is important to show the staining for astrocytes early on.

Reply:

We thank the Reviewer for the suggestions as they greatly improve the paper. In the revised manuscript we have now included new Sholl analysis performed for GFAP⁺ astrocytes in the SNpc and striatum of 4-month-old rats, as well as in the hippocampus and pontine nuclei (**new Fig. 4, Supplementary Fig. 3**). Also, in line with the request of Reviewer 3 (point 1), the morphology data on microglia and astrocytes is now combined with a more accurate, stereological counting of cell numbers (**new Fig. 4 and Supplementary Fig. 3,4**: these data substitute the ones shown in the old Suppl. Fig. 2e). Overall, no changes are observed in either astrocyte numbers or morphology in any of the areas examined.

Points 9-10 raised by Reviewer #1:

-The authors should address if areas outside the basal ganglia show a similar increase in microglia numbers or change in morphology.

- Similarly, it is unclear if the observed microglial activation is region-specific and if that correlates with the expression of alpha-synuclein.

Reply:

In line with the Reviewer's suggestion, additionally to the SNpc and striatum we've analysed the morphology and stereological cell numbers of microglia (and astrocytes) in the dorsal hippocampus and pontine nuclei. The numbers of Iba+ cells were indeed increased in the hippocampus of Syn rats, and the Sholl analysis showed similar change in morphology as what we observed in the SNpc and striatum (**new Supplementary Fig. 4**), while the morphology changes were prevented after a two-month treatment with RvD1 (**Supplementary Fig. 6 and 7**).

On the other hand, changes in microglia were not seen in the pontine nuclei (**new Supplementary Fig. 4**); it is unclear if this difference correlates with the expression of alpha-synuclein: the paper by Nuber et al., (Brain. 2013 ;136(Pt 2):412-32), from the group that first developed and characterised the Syn rat model, shows that the a-syn levels in 3-month old rats are very low in the SNpc (compared to other brain areas like the striatum or hippocampus), yet the microglial activation we observed in the SNpc is as strong as that observed in these other regions (with a near 2-fold increase in Iba cell numbers).

Point 11 raised by Reviewer #1:

- The authors state that there is a relationship between alpha-synuclein, microglia, and pro-inflammatory cytokines and suggest that this interaction occurs early on and that mechanisms driving resolution of inflammation may be at fault. However, these potential mechanisms are not discussed.

Reply:

The fact that the observance of several neuroinflammatory events (i.e. microglia activation, brain IFN- γ expression) is temporally associated with alterations in the levels of RvD1 only in those animals overexpressing α -synuclein, suggests that these phenomena are linked. This is corroborated by our reinstating the Syn animals with RvD1, whose chronic treatments restored all neuroinflammatory processes. We now make an attempt in the Discussion to discuss some potential mechanisms of action of resolvins for halting inflammation.

Point 12 raised by Reviewer #1:

- To promote the translation value of RvD1 treatment, the authors should thoroughly examine the pharmacokinetics and bioavailability of RvD1 in rats/humans. It is unclear how the treatment concentration was chosen. To provide rigor, it is important to show how neuroinflammatory signaling cascades are perturbed/restored with different experimental conditions (i.e., WT vs Syn +/- RvD1).

Reply:

We agree with the Reviewer and we performed a pharmacokinetics analysis of RvD1 (**new Supplementary Fig. 5b**) in order to also examine its bioavailability. As a matter of fact, both bioavailability in plasma and the concentration used for in vivo studies has been extensively reported in many studies (for chronic administrations the used RvD1 concentration is always between 0.1 and 0.2 µg/kg and given every other day or twice a week, whereas for acute single injections it is usually used up to 0.4-0.5 µg/kg (Recchiuti et al., FASEB J 2014; Hsiao HM et al., Am J Pathol 2015; Poisson LM et al., J Biol Chem 2015; Chiurchiù et al., Sci Transl Med 2016; Norling LV et al., JCI Insight 2016; Bisicchia et al., Mol Neurobiol 2018; Sun RA et al., Sci Rep 2019).

As suggested, in order to better show how neuroinflammatory events are perturbed/restored in the different experimental groups and upon RvD1 treatment, we've also performed a more thorough analysis of immune cells and inflammatory markers by means of polychromatic flow cytometry, as shown in the new **Fig. 4 and Fig.6 and Supplementary Fig. 5**.

Point 13 raised by Reviewer #1:

-The patient cohort included in this study ranges from 50 to 80 years old. It is unclear how that compares to the 4-month-old rats. Importantly, more clinical information, such as the UPDRS rating of patients should be reported.

Reply:

Although one of the criteria for the patient recruitment was to exclude under-50s or over-80s patients, the actual age range for PD patients was 52-73 years of age (42-78 for control subjects). Overall, we've limited our study on very mildly affected patients (defined as early-PD in the text) not exceeding stage 2 in the H&Y scale, which, we believe, could match the early motor symptoms observed in 4-month-old rats. Nonetheless, due to the complexity of the human disease, we've modified the text trying to avoid direct comparison of data from humans and rats. Additionally, the clinical data from individual PD patients are now presented in **Supplementary Table 1**.

Minor Points raised by Reviewer #1:

1. The authors state in the title that RvD1 prevents neuroinflammation and thereby onset of PD. However, they state in the discussion (line 554) that it delays the onset of PD. Based on the data shown, 'prevents' would seem more appropriate. Without looking at the disease progression more systematically/thoroughly, the word "delays" is confusing.

Reply:

This is now corrected in the text.

2. Line 42-43 is a confusing sentence (family of omega 3 fatty acids-derived specialized pro-resolving mediators that include resolvins).

Reply:

The Abstract is now changed, to comply with the journal's format and with the new data. This sentence is now changed.

3. The reconstructed cells should have the soma filled with a darker shade so as to help the readers.

Reply:

The figures are changed as asked. This was possible only for Iba cells but not GFAP cells, as the soma of astrocytes is particularly small.

4. The authors should not assume that all data have a normal distribution and should determine if non-parametric tests are needed.

Reply:

We performed normalization tests for all data using the D'Agostino&Pearson and Shapiro-Wilk tests and revised all statistical analysis in the main text and supplementary files. The Statistics paragraph in Methods and all figure legends are now changed accordingly. In the text we also now included the P values for non-significant data.

5. The authors should state if the Syn rats are from Sprague-Dawley background.

Reply:

Yes, Syn rats have a Sprague-Dawley background. This is now added in the animal section (Methods).

6. The authors should explain why the kinetics of baclofen elicited currents are different from that of DA-induced currents.

Reply:

This is likely due to the fact that the concentration of baclofen in the original traces (1 μ M) is very low, near EC15 for rat DA neurons (Cruz et al., 2004; Nat Neurosci. 2004;7(2):153-9), whereas the DA concentration is near-saturating. In the revised manuscript (**Fig. 3c**) we now show responses to 10 μ M baclofen (~EC40).

POINT-BY-POINT REPLY TO REVIEWER #2

Reviewer #2:

This study investigates the role for resolving D1 (RvD1) in a transgenic rat model of Parkinson disease and provides compelling evidence that RvD1 is involved in the disease activity. The authors provide detailed characterization of the relatively new rat model based on transgenic expression of alpha-synuclein, including data on behavior (motor and non-motor), neuron loss, and dopaminergic neuron function. In this model, they find evidence for microglial activation and altered levels of RvD1 in both plasma and CSF, and report that both the physiological abnormalities and the morphological activation of microglia are rescued by administration of RvD1. This work is strengthened by the inclusion of human data, which demonstrates that RvD1 is decreased in the CSF and plasma of early Parkinson patients. Overall, this article provides evidence that resolvins are important in the pathogenesis of Parkinson disease, a novel idea worthy of further study.

There are some issues that should be addressed.

Point 1 raised by Reviewer #2:

1) The rat model, while thoroughly characterized at the behavioural and electrophysiological levels, does not show neurodegeneration at early time points. Authors cite a paper that states neurodegeneration occurs later in the model, but the citation is incorrect and references a review article by Ransohoff et al.

Reply:

The Reviewer is quite correct, and we apologise for the mistake. The correct reference is Nuber et al., (Brain. 2013 ;136(Pt 2):412-32), from the group that first developed and characterised the Syn rat model. In this paper the authors showed impaired DAT-scan at 16 months of age and DA neuron neurodegeneration at 18 months of age. All references are now corrected in the text.

Point 2 raised by Reviewer #2:

2) The authors do not specify how the dose of RvD1 was determined. Authors should provide a clear rationale and pharmacological data for dosing or cite a relevant paper if the dosing has already been established in the literature.

Reply:

We apologise for not addressing this issue before; we now provide a rationale for the choice of RvD1 concentration in the Results section by citing some relevant papers by our own group that previously determined the efficacious concentration of RvD1 and the modality of treatment in acute

and chronic conditions. We also provide a pharmacokinetic analysis of RvD1 in the plasma following single treatment (**new Supplementary Fig. 5b**). Please see also Reply to Reviewer 1, point 12.

Point 3 raised by Reviewer #2:

3) In Figures 6 and 7, the authors administer RvD1 to correct the various phenotypes found in the rat model. However, it would be important to determine whether this treatment in fact corrects the deficiencies of RvD1 in CSF, plasma and brain.

Reply:

In line with the Reviewer's request, we have now performed measurements of RvD1 in CSF and plasma of WT and Syn rats following a new 2-month treatment. Indeed, the treatment increases the levels of RvD1 in both CSF and plasma (**new Supplementary Fig. 5c,d**). Additionally, we also investigated the ability of RvD1 treatment to correct deficiencies in the number of leukocytes in the periphery. These data are shown in **new Fig. 4 and 6** (for naïve and treated rats, respectively) and **Supplementary Fig. 5e**. See also response to Point 5 raised by Reviewer #3.

Point 4 raised by Reviewer #2:

4) Finally, the authors use bar graphs that display each data point within the bar. However, the number of data points do not always match the reported number of animals used. It is possible that multiple animals fall at the same point on the graph-authors should use a graph format that displays similar data points side by side.

Reply:

Indeed, some data points overlapped in the plots; all figures are now changed to show data in a side-by-side arrangement.

Point 5 raised by Reviewer #2:

5) It seems that the authors have mixed up which figures they are referring to when discussing Figures 5 c, d in the text. Also, Figure 5e is never mentioned in the text.

Reply:

We apologize for the confusion. The text regarding **Figure 5** is now corrected.

POINT-BY-POINT REPLY TO REVIEWER #3

Reviewer #3:

In the manuscript Cordella et al. approach early changes in a Parkinson's disease (PD) model based on human α -synuclein (α -syn) expression in rats, and the potential of resolvin D1 (RvD1) as a neuroprotective drug that can resolve neuroinflammatory changes associated to α -syn and protect neurons.

The BAC human α -syn rats, express the protein under the human promoter. These rats have been previously shown to exhibit certain dopaminergic deficits and α -syn pathology, although these changes were observed at later age. In the manuscript, the earliest signs of change are observed at 4 months, so authors selected 2 and 4 months as time points for the study. The authors describe early (4 months) changes in behavior and in the firing pattern of dopaminergic (DAergic) nigral cells that were associated to differences in DA release in striatum and changes in DA sensitivity possible related to GIRK/Kir3 channels. This was observed in the in the absence of any cellular (in substantia nigra (SN)) or axonal (in striatum) DAergic loss.

The authors relate these changes to early microglia response in SN and striatum, as shown by increase in the number of microglia and in their morphological complexity. In addition, they report increased IFN γ in CSF. Interestingly, these inflammatory changes were associated to elevated RvD1 in CSF, while the plasma level of RvD1 was decreased, suggesting an opposing event occurring in periphery and in brain at early stages. However at 18 months of age, the RvD1 was decreased both in CSF and plasma. Interestingly a similar decrease in RvD1 is also found in CSF and plasma from PD patients.

The team attempts to examine if treatment with RvD1 can result in neuroprotection. After 2 months of treatment all reported changes were abolished and the animals appeared similar to WT, suggesting a therapeutic ability for the RvD1 in α -syn induced neurodegeneration and inflammation.

While the early neurodegenerative signs described in the model and its correlation with immune response is relevant, and the protection with RvD1 very promising, several points need to be addressed in the work.

Point 1 raised by Reviewer #3:

1. Although the quantification of microglia morphological profile is adequate and nicely presented, the excessively simplistic approach for the quantification of number of microglia (and astroglia) in SN and in striatum is very poor. The authors should approach a stereological quantification that more accurately can give a realistic estimation of total number of microglia through the two brain areas: SN and striatum.

Reply:

In line with the request of Reviewer 3, the morphology data on microglia and astrocytes are now accompanied by stereological counting of GFAP- and Iba1-positive cell numbers in the SNpc and striatum (**new Fig. 4, Supplementary Fig. 3 and 4**). We've also performed a stereological cell counting and morphological analysis of GFAP- and Iba1-positive cells in the hippocampus and pontine nuclei (**Supplementary Fig. 3 and 4**) and increased the sample size for the morphology data of microglia in the SNpc and striatum (see reply to Minor point 1). The Methods and Results sections are also changed accordingly.

Point 2 raised by Reviewer #3:

2. In addition, the analysis of other inflammatory markers should have brought some light to the study. Such as MHCII, CD68, TLR2 or TLR4 expression, monocyte and lymphocytes infiltration in striatum or SN.

Reply:

In an attempt to make a more thorough investigation of inflammatory changes in Syn rats, we examined the percentage of several immune cell (granulocytes, monocytes and lymphocytes) populations in naïve animals (**new Fig. 4**); given that in Syn rats we observed a decrease in the population of monocytes, we also examined the immunophenotype of these cells for MHCII, CD68 and investigated whether the RvD1 treatment could restore the observed increases in these markers (**new Fig. 6, Supplementary Fig. 5e**).

Point 3 raised by Reviewer #3:

3. Also are these changes observed found only for dopaminergic nigro-striatal areas or where other areas in brain also showing signs of inflammation?

Reply:

Other than the SNpc and striatum, we investigated the morphology and stereological cell numbers of GFAP- and Iba1-positive cells in the dorsal hippocampus and pontine nuclei (**new Fig. 4; Supplementary Fig. 3 and 4**). We saw that the numbers of Iba+ cells were increased in the

hippocampus of Syn rats, and the Sholl analysis showed similar change in morphology as what we observed in the SNpc and striatum. On the other hand, these changes were not seen in the pontine nuclei. GFAP cells were unchanged in all four regions. See also Reply to Comments 8-10 of Reviewer 1.

Point 4 raised by Reviewer #3:

4. The authors observed change in RvD1 in peripheral as well as in brain. The literature suggest changes in the immune system in PD includes also peripheral immune changes. However, the authors do not address whether the observed changes are due to peripheral immune events (except for the selected panel in plasm) that might ultimately reflect in brain changes. In general, I think the study requires a better characterization of immune response, in brain and in periphery. This seems especially relevant based on the opposite changes observed in RvD1 at early ages with increase in CSF while decrease in plasma.

Reply:

We thank the Reviewer for this comment. As a matter of fact, most of the literature on PD does not report significant changes in the immune system and in pro-inflammatory cytokines in the periphery. Most cytokines for example are undetectable in plasma of PD patients and in animal models, and if any are detected, the levels are in the range of 1-10 pg/ml, which is considered to be very low. Even changes in blood leukocytes are usually discordant, with several papers showing no change and few ones showing either small, yet significant, increases or decreases in specific cell types (Delgado-Alvarado et al., *Mov Disord* 2017; Rocha et al., *Mol Neurobiol* 2018), making this issue a much debated one. In general, PD is not usually associated to a peripheral immune disease (unlike, for example, multiple sclerosis) but instead is strongly linked to neuroinflammation of resident immune cells, e.g. microglia. Indeed, no particular evidence of immune cell infiltrations has still been reported in the brain of PD patients or animal models.

In line with the Reviewer's suggestion, in the revised manuscript we investigate peripheral immune changes in our Syn model by examining the percentage of leukocyte populations (neutrophils, T lymphocytes, monocytes, etc.) and their immunophenotype in 4-month-old naïve WT and Syn animals, and also in animals that underwent the 2-month RvD1 treatment. These results are included in the **new Fig. 4 and 6** and in **Supplementary Fig. 5e** (for naïve and treated rats, respectively).

Point 5 raised by Reviewer #3:

5. How were the RsD1 levels in brain and plasma after the 2-month RsD1 treatment?

Reply:

We thank the Reviewer for this comment. We analyzed the RvD1 levels in CSF and plasma following a new 2-month treatment with RvD1 and found a significant increase in both tissues, suggesting that the chronic treatment is indeed reflected in a boost of its levels in both plasma and brain. This result is included in the **new Supplementary Fig. 5c,d**. See also response to Point 3 raised by Reviewer #2.

Point 6 raised by Reviewer #3:

6. Analysis of pathological a-syn should also be included with markers such as Phospho-a-syn and antibodies such as MJF14 (please see the MJ Fox Foundation resources web-page) at both ages and treatments. Are these changes associated to detectable aggregated or pathological a-syn or were they prior to any detectable pathology?

Reply:

We did not perform an analysis of pathological a-syn since a very thorough investigation of a-syn aggregates in this rat model has been recently published by Nuber et. al., who first developed this model (Brain 2013: 136; 412–432). According to this paper, pathological changes such as insoluble a-syn aggregates or dystrophic nerve fibres are not present in 3-month old rats but in much older animals. Additionally, the changes we observe are prior to any detectable pathology in terms of dopamine neuron degeneration or denervation both in 2 and 4 months of age (see **Fig.1**). We've rephrased the last paragraph referring to Fig. 1 (pg. 6) to make this point clearer.

Point 7 raised by Reviewer #3:

7. The reference list includes 109 references although in the text only 75 are cited. In fact the citation given in text never corresponded to the article with such number in the reference list. This made the reading process extremely complicated.

Reply:

The Reviewer is quite correct, and we apologise for the mistake. All references are now corrected in the text.

Point 8 raised by Reviewer #3:

8. The discussion regarding the putative role of resolvin in the model and the protection could be improved (although due to the chaotic reference list, it was difficult to follow). The authors should reflect or suggest putative mechanism. Is this effect of RvD1 a consequence of its effect in neurons or in microglia? The reference regarding the RvD1 protection in the MPP+ model (in text indicated

as #74 and in reference list #107) suggests an effect on neurons directly. However, the RsD2 in the LPS model seems to exert the protection through the microglia (reference in text indicated as #75 and in reference list #106). RsD have been related to inhibition of a TLR signaling through glycogen synthase kinase 3 (GSK3) (PMID:26878867 PMID:16007092) and also a-syn activation of microglia seems related to TLR4 (PMID:23108585) and TLR2; and furthermore, TLR2 seems also relevant in neurons in PD (PMID:27888296). Resolvins have been also associated to T-cell activation (PMID:27559094), and T-cells have been related to PD and a-syn (PMID:26018603). In that regard, IFN-g is an important mediator of T-cell response.

Reply:

We agree with the Reviewer; In the revised manuscript we make an attempt to provide a discussion of the potential mechanism of our findings, also in the light of the current literature on RvD1.

Minor Points raised by Reviewer #3:

1. In page 10 the authors indicate ‘15 cells per animals and per experimental group were selected...’ However, little is told about how the data is handled later. Were the markers studied averaged per animal and subsequently per group?

Based on fig 4 only 3 animals were used. Despite the differences found this is very small number, in fact this is true for many of the measurements which has 4 or less animals used.

Reply:

Yes, all 15 cells were averaged per animal; subsequently all animals were averaged per experimental group (for the plots showing relationship to radial distance from soma) or shown as individual points (for plots showing soma area and perimeter). The Methods section is changed to make this clearer to the reader. Additionally, we analysed more animals in order to increase the sample size for the Scholl analysis (**Fig. 4 and Supplementary Fig. 3,4**), and also increased the sample size for RvD1, RvD2 and INF- γ measurements in naïve and treated rats following a new 2-month treatment (**Fig. 4g; Fig. 5b,c; Fig. 6f**).

2. In graphs with individual numbers represented by points, they should be drawn in a way that two points of equal value can be distinguished (i.e distributed not only vertically but also horizontally). In that regard, please in fig 8 indicate out of the total n number how many patients showed detectable levels of each marker, since in the graphs this is not possible to really asses.

Reply:

All figures are now changed to show data points in a side-by-side arrangement. The data in **Fig. 8** are now more legible to show the number of patients with detectable cytokines.

3. In figure 6F the authors show that CSF IFN γ in wt saline is significantly different than a-syn saline, but based on the graph that seems unlikely. However, the graph suggest that the treatment leads to decrease of IFN γ in the WT animals.

Reply:

We've now increased the number of all the analysed animals to 7 and re-performed the statistical analysis to increase the efficacy of the ANOVA test and of the post-hoc comparisons. The new data now show a clear difference between the WT-sal and Syn-sal rats (in line also with what occurs in 4-month old naïve animals).

The treatment does have an effect in both WT and Syn rats (in line with the anti-inflammatory nature of RvD1), but importantly, following treatment the difference between WT and Syn rats is cancelled (please compare WT-RvD1 with Syn-RvD1). We've now changed the text to make this clearer to the reader.

4. Page 20 In the text describing the changes of RvD1 and RvD2 with age, the authors should indicate: compare Fig5b vs. 5d and 5c vs.5e, to help the reader.

Reply:

As suggested, we've made changes in the text describing **Fig. 5**, that hopefully make it easier for the reader. See also reply to Point 5 raised by Reviewer #2.

5. It is unclear to this reviewer whether the statistical approach used is strictly correct or not. The authors indicate that they approach each factor as independent variable, irrespective that the factors show interaction, which does not seem correct. A statistic expert should comment in that particular point.

Reply:

The main purpose of the 2-way ANOVA test is to understand if there is interaction between two independent variables (such as genotype and treatment) on a dependent variable (such as time spent on the rotarod tube). Indeed, in cases of significant interaction between the two independent variables, on its own the ANOVA test should not be used to draw conclusions from the experiment other than simple interpretations such as 'the motor performance *is affected* by both genotype and treatment' (i.e. the test does not provide information as to *how* the motor performance is affected). In cases of interaction, to draw more meaningful conclusions one has to accompany the 2-way ANOVA test with additional tests such as 1-way ANOVAs for each independent variable, or more commonly-used post-hoc tests that essentially break the interaction effect into component parts and

then test the separate parts for significance. All our 2-way ANOVA tests were followed by Bonferroni's post-hoc tests and the individual comparisons are described in the legends and figures (shown as asterisks). Of note, there was no difference for any experiment between Bonferroni's and Tukey's tests.

Additionally, as asked by Reviewer #1, we revised all the statistics in the main text and supplementary file, performed normalization tests for all data and changed the Figure legends and Statistics paragraph accordingly. In the text we've also now included the P values for non-significant data.

Reviewers' comments:

Reviewer #1 (Remarks to the Author):

This paper has been thoroughly revised. The authors have sufficiently addressed my concerns along with those raised by the other two reviewers. I do not have additional questions/comments at this point. The study overall is excellent and timely. It should be published without further delays.

Reviewer #2 (Remarks to the Author):

On the whole, this manuscript is much improved. The authors have added additional data, and have addressed most of the points raised previously. I have only some minor suggestions for improvement:

1) The finding that the monocytes in the blood of the rat model are reduced (pg 11) is interesting, but the explanation offered on page 17 seems unlikely. The authors suggest that the blood monocytes are reduced because they have infiltrated the brain. While there may indeed be infiltrating monocytes in the brain, it is not plausible that this is the cause of the reduced numbers in the blood. Do they know what happens to the blood monocyte numbers after treatment with RvD1?

2) The comparative studies of RvD1 and RvD2 in rats and humans are valuable, but the Discussion does not really address the differences in these findings directly. It would be helpful to be more specific about this.

3) Most of the values in Table 1 are either "not detected" or not statistically significant. This table could be condensed or eliminated, and the significant values simply reported in the text.

4) In figure 8, most of the data displayed are not statistically significant. The most important panels are in C and D, illustrating the resolvins. Many of the other panels could be eliminated.

Reviewer #3 (Remarks to the Author):

The authors have addressed most of the points asked. However, a couple of points are yet to be considered:

In their answer to "Point 4 raised by Reviewer #3:"

I disagree with the statement that there is little evidence of the peripheral inflammatory component in PD. Several labs have studied the changes in monocytes and found alterations in patients' blood immune cells (Fiszer et al., 1994; Bessler et al., 1999; Hasegawa et al., 2000; Reale et al., 2009; Luo et al., 2010; Funk et al., 2013; Grozdanov et al., 2014; Drouin-Ouellet et al., 2015; Bliederhaeuser et al., 2016; Schlachetzki et al., 2018; Smith et al., 2018; Wijeyekoon et al., 2018). Similarly studies analyzing T-cells have also shown changes in the adaptive immune system (Fiszer et al., 1994; Bas et al., 2001; Hisanaga et al., 2001; Baba et al., 2005; Gruden et al., 2011; Niwa et al., 2012; Saunders et al., 2012; Stevens et al., 2012; Sulzer et al., 2017; Kustrimovic et al., 2018; Rocha et al., 2018). As well as B-cells (Bas et al., 2001; Gruden et al., 2011; Stevens et al., 2012; Kobo et al., 2016).

Also different groups have addressed monocytes infiltration in a-syn based models (Harms et al., 2017; Harms et al., 2018) and the influence of peripheral immune cells on the microglia response

to a-syn-related events in periphery or in CNS (Sanchez-Guajardo et al., 2015; Peralta Ramos et al., 2019).

Indeed, this is confirmed by their own observations of changes in monocytes. Notice, that the lack of changes in the other immune cells, might reflect a true absence of differences, or might be only a consequence of the superficial analysis performed that relied in only one (and a second excluding) marker per population. The only exception being monocytes, which when approached more in detail (using MHCII and CD68), showed changes. Thus this point should be discussed in the manuscript.

Please delete the statements of line 245-246 regarding a "thorough" characterization. The authors have performed a gross analysis of total number of T-cells, B-cells and granulocytes, and a more detailed analysis of the monocyte/macrophages (using MHCII and CD68 in the CD11b+ cells.)

Equally in lines 398-401, please rephrase. Although there were no significant changes in the percentage of T-cells, B-cells and Granulocytes in blood, possible functional changes in these populations were not addressed and thus cannot be discarded.

The authors find changes in RvD1 in brain and periphery as well as changes in brain microglia and peripheral monocytes after treatment. Based on these findings, the authors cannot discard that the effects seen are a summary or a synergistic event of both, peripheral as well as CNS immune response. Thus, this aspect needs to be discussed as a plausible factor more clearly and not being discarded so quickly in the manuscript.

Regarding flow cytometry: Please provide information on the use of compensation and FMO controls; as well as specific antibody clones used. Furthermore, a figure showing the full gating strategy and not only the FSC/SSC plot should be included. Did the authors use a live/dead marker? Were all markers used in multiple color panels, or were several panels run in parallel? The sentence "data expressed as % of cells positive for the given fluorochrome-conjugated antibodies " is not clear. Is the % of the parent population (e.g. lymphocytes) or total (live) cells?

In their answer to Point 6 raised by Reviewer #3:

I also disagree with the authors in this point and statements along the paper: like lines 128-129. Nuber et al., showed in their original paper that some pathological aggregation of a-syn and dystrophy of striatal axons is already apparent t 3 months in the rats (fig 1B in the original Nuber's publication). Similarly, other post mortem protein analysis by western blots in the paper of 2013 already showed changes in a-syn oligomerization at 3-months in some brain areas, such as cortex (main input of the striatum). Although the analysis in the 2013 paper was rather detailed at the moment, it highly relied on western blots, proteinase K and antibodies not suitable for histology, such as the Jensen's antibody FILA. And this approach might miss discrete early changes. Since then, new tools have been developed that can be used for the histological evaluation of early a-synuclein pathology as the antibodies suggested in my prior review (MJF14 for example). The importance of the selective and early immune alterations in the different areas of the brain tested by the authors, makes highly relevant the questions, whether the local pathological accumulation of a-synuclein is concomitant with the immune changes or not. Is microglia responding to early changes that precedes the pathological accumulation ? Are they activated by undetectable forms of modified a-synuclein release by the neurons or can we detect already such a-synuclein species? Or on the other hand, are microglia activated by the functional changes of the neurons, that will be also sensed by microglia (since they possess most of the neurotransmitter receptors)?

The argument written by the authors stating that clear signs of degeneration are not seen and therefore no a-synuclein pathology should be found, is not valid, since the current hypothesis is that a-synuclein pathological misfolding and modification will occur prior to any apparent degeneration, but with functional changes (as seen here).

In order to answer (at least partially) these questions, the authors should perform the immunostaining in sections containing: SN, Striatum, hippocampus and pontine nucleus (and cortex as positive control(?)) with the antibodies suggested or alternative relevant ones.

Additional points:

1. Regarding the stereological quantification of the number of Iba1+ cells, how is possible that the total number of microglia in Figure 4B in striatum is 5000 and 10000, while in Fig 6, it is in the range of 50.000 and 100.000? (possibly a mistake in the graphs).

2. Interestingly the treatment, did not change microglia in hippocampus. Do the authors have any hypothesis of such observation? This could be highly relevant, since the rat BAC line, has been shown to have defects in hippocampal neurogenesis at this early time point, which could also be related to the microglia activation in the area. Thus , the treatment might not improve some non-motor problems in patients?

We would like to thank the Reviewers for their comments, we hope that these are now satisfactorily addressed in the revised version of the paper. Below is a point-to-point reply, while in the main text all the relevant changes are highlighted for easiness of reading. In line with the journal's policy, we also provide the revised checklists regarding data analysis, and the revised 'Source Data file' with all the raw data and full blot.

Reviewer #1 (Remarks to the Author):

This paper has been thoroughly revised. The authors have sufficiently addressed my concerns along with those raised by the other two reviewers. I do not have additional questions/comments at this point. The study overall is excellent and timely. It should be published without further delays.

Reply: We thank the Reviewer for his/her appreciation and help in improving our manuscript.

Reviewer #2 (Remarks to the Author):

On the whole, this manuscript is much improved. The authors have added additional data, and have addressed most of the points raised previously. I have only some minor suggestions for improvement:

1) The finding that the monocytes in the blood of the rat model are reduced (pg 11) is interesting, but the explanation offered on page 17 seems unlikely. The authors suggest that the blood monocytes are reduced because they have infiltrated the brain. While there may indeed be infiltrating monocytes in the brain, it is not plausible that this is the cause of the reduced numbers in the blood. Do they know what happens to the blood monocyte numbers after treatment with RvD1?

Reply: We thank the Reviewer for making this point and allowing us to clarify better. The infiltration of monocytes in the brain, as an explanation of the reduced levels of monocytes in the blood, is a possibility given that we observed an increase of Iba1+ cells in the brain, which is also a marker of infiltrated monocytes/macrophages (nowadays it is difficult to distinguish between these two different cell types), but also given the evidence from other studies showing brain infiltration of monocytes/macrophages occurring in PD (Harms et al., 2017; Harms et al., 2018; Moehle and West, 2015). This, of course, is just a hypothesis and doesn't rule out other possible explanations. For example, the reduced number of blood monocytes could be due to an alteration of the

inflammatory responses occurring in the periphery, or due to cell death following inflammatory activation, or due to re-mobilization from primary and secondary lymphoid organs. These possibilities have now been added in the Discussion (line 437).

Regarding the effect of RvD1 treatment on blood monocytes number, we had already shown this (see **Fig. 7g,h**); treatment with the pro-resolving lipid significantly restored monocyte blood numbers in Syn rats and reduced their activation state. See also **Supplementary Fig. 6b** for the effect of RvD1 on other peripheral blood populations.

2) The comparative studies of RvD1 and RvD2 in rats and humans are valuable, but the Discussion does not really address the differences in these findings directly. It would be helpful to be more specific about this.

Reply: We better stressed this issue in the Discussion (see line 487).

3) Most of the values in Table 1 are either “not detected” or not statistically significant. This table could be condensed or eliminated, and the significant values simply reported in the text.

Reply: Table 1 is amended, as suggested.

4) In figure 8, most of the data displayed are not statistically significant. The most important panels are in C and D, illustrating the resolvins. Many of the other panels could be eliminated.

Reply: Unlike for Table 1, where out of all the 7 measured cytokines only 2 were detected in the rat, here we believe that it is important to show all cytokine levels in patients because both in the CSF and blood all cytokines were indeed detected, with some of them showing high levels (IFN-g, IL-4, IL-13). The fact that most of them are not statistically significant is still an important result to show, especially compared to the statistically significant variation of RvD1.

Reviewer #3 (Remarks to the Author):

The authors have addressed most of the points asked. However, a couple of points are yet to be considered:

1. In their answer to ”Point 4 raised by Reviewer #3:”

I disagree with the statement that there is little evidence of the peripheral inflammatory component in PD. Several labs have studied the changes in monocytes and found alterations in patients' blood immune cells (Fischer et al., 1994; Bessler et al., 1999; Hasegawa et al., 2000; Reale et al., 2009; Luo et al., 2010; Funk et al., 2013; Grozdanov et al., 2014; Drouin-Ouellet et al., 2015; Bliederaeuser et al., 2016; Schlachetzki et al., 2018; Smith et al., 2018; Wijeyekoon et al., 2018). Similarly studies analyzing T-cells have also shown changes in the adaptive immune system (Fischer et al., 1994; Bas et al., 2001; Hisanaga et al., 2001; Baba et al., 2005; Gruden et al., 2011; Niwa et al., 2012; Saunders et al., 2012; Stevens et al., 2012; Sulzer et al., 2017; Kustrimovic et al., 2018; Rocha et al., 2018). As well as B-cells (Bas et al., 2001; Gruden et al., 2011; Stevens et al., 2012; Kobo et al., 2016). Also different groups have addressed monocytes infiltration in a-syn based models (Harms et al., 2017; Harms et al., 2018) and the influence of peripheral immune cells on the microglia response to a-syn-related events in periphery or in CNS (Sanchez-Guajardo et al., 2015; Peralta Ramos et al., 2019). Indeed, this is confirmed by their own observations of changes in monocytes. Notice, that the lack of changes in the other immune cells, might reflect a true absence of differences, or might be only a consequence of the superficial analysis performed that relied in only one (and a second excluding) marker per population. The only exception being monocytes, which when approached more in detail (using MHCII and CD68), showed changes. Thus this point should be discussed in the manuscript.

Reply: We apologize for minimizing this issue in our previous reply. We completely agree with the reviewer on the fact that peripheral immune changes have indeed been observed in PD in many studies, and the evidence that alterations of several immune cell populations indicates that peripheral immunity plays a critical role. This issue is now better discussed in the text, in view of our own results regarding changes in monocytes, and some of the above-mentioned works have been cited in the revised manuscript. As for the lack of changes of other peripheral immune cells, we provided a possible explanation in the Discussion, as suggested (see line 441).

2. Please delete the statements of line 245-246 regarding a “thorough” characterization. The authors have performed a gross analysis of total number of T-cells, B-cells and granulocytes, and a more detailed analysis of the monocyte/macrophages (using MHCII and CD68 in the CD11b+ cells.)

Reply: The phrase has been amended, as suggested (line 249); see also line 324 where we specify that the RvD1 treatment does not have an effect *in the numbers* of other cell populations.

3. Equally in lines 398-401, please rephrase. Although there were no significant changes in the percentage of T-cells, B-cells and Granulocytes in blood, possible functional changes in these populations were not addressed and thus cannot be discarded. The authors find changes in RvD1 in brain and periphery as well as changes in brain microglia and peripheral monocytes after treatment. Based on these findings, the authors cannot discard that the effects seen are a summary or a synergistic event of both, peripheral as well as CNS immune response. Thus, this aspect needs to be discussed as a plausible factor more clearly and not being discarded so quickly in the manuscript.

Reply: We agree with the Reviewer; we better discussed this in the revised manuscript.

4. Regarding flow cytometry: Please provide information on the use of compensation and FMO controls; as well as specific antibody clones used. Furthermore, a figure showing the full gating strategy and not only the FSC/SSC plot should be included. Did the authors use a live/dead marker? Were all markers used in multiple color panels, or were several panels run in parallel? The sentence “data expressed as % of cells positive for the given fluorochrome-conjugated antibodies” is not clear. Is the % of the parent population (e.g. lymphocytes) or total (live) cells?

Reply: The requested information has been added in the **Methods**. As for the gating strategy, we apologize if it was not clear enough in the Methods, this is now corrected. Briefly, cells were simultaneously stained with a single mix containing all the 6 different antibodies in different colors (anti-granulocytes-FITC, CD3-PE-Vio770, CD45RA-PE, CD11b/c-APC, MHC-II-VioBlue and CD68-APC-Vio770) and acquired on a 10 color-flow cytometer (Cytotflex, Beckman Coulter). Total leukocytes were gated and after excluding both cell doublets and eventual dead cells (using LIVE DEAD Zombie Aqua dye), the % of cells expressing either anti-granulocytes and CD11b (monocytes) were plotted. Inside the CD11b⁻ population, we plotted CD3 and CD45RA for the identification of T and B cells. CD11b⁺ monocytes were further gated to observe the % expression of MHC-II and CD68. The procedure is now described in **new Supplementary Fig. 6a**. More information can also be found in the **Reporting Summary** that accompanies the manuscript.

5. In their answer to Point 6 raised by Reviewer #3: I also disagree with the authors in this point and statements along the paper: like lines 128-129. Nuber et al., showed in their original paper that some pathological aggregation of a-syn and dystrophy of striatal axons is already apparent at 3 months in the rats (fig 1B in the original Nuber's publication). Similarly, other post mortem protein analysis by western blots in the paper of 2013 already showed changes in a-syn oligomerization at

3-months in some brain areas, such as cortex (main input of the striatum). Although the analysis in the 2013 paper was rather detailed at the moment, it highly relied on western blots, proteinase K and antibodies not suitable for histology, such as the Jensen's antibody FILA. And this approach might miss discrete early changes. Since then, new tools have been developed that can be used for the histological evaluation of early a-synuclein pathology as the antibodies suggested in my prior review (MJF14 for example). The importance of the selective and early immune alterations in the different areas of the brain tested by the authors, makes highly relevant the questions, whether the local pathological accumulation of a-synuclein is concomitant with the immune changes or not. Is microglia responding to early changes that precedes the pathological accumulation? Are they activated by undetectable forms of modified a-synuclein release by the neurons or can we detect already such a-synuclein species? Or on the other hand, are microglia activated by the functional changes of the neurons, that will be also sensed by microglia (since they possess most of the neurotransmitter receptors)?

The argument written by the authors stating that clear signs of degeneration are not seen and therefore no a-synuclein pathology should be found, is not valid, since the current hypothesis is that a-synuclein pathological misfolding and modification will occur prior to any apparent degeneration, but with functional changes (as seen here).

In order to answer (at least partially) these questions, the authors should perform the immunostaining in sections containing: SN, Striatum, hippocampus and pontine nucleus (and cortex as positive control(?)) with the antibodies suggested or alternative relevant ones.

Reply: We thank the Referee for these pivotal questions, as they are essential for understand better the disease pathology. As the Reviewer suggested, we used the conformation-specific MJF14 antibody to analyse the formation of a-Syn aggregates in the midbrain, cortex, hippocampus, pontine nuclei and striatum. A quantitative assessment of a-Syn fibrils using dot blots showed that all brain areas tested, except for the pontine nuclei, show significantly increased immunoreactivity for the MJF14 antibody in 4-month-old Syn rats compared to controls. These data are shown **in new Fig. 5**, and the text is changed accordingly (also lines 128-129 are now omitted). Of note, this result is in line with our data showing reactive microglia in the midbrain, hippocampus and striatum, but not in pontine nuclei. Thus, it appears that there is indeed correlation between a-syn aggregation and microglia activation. Unfortunately, this set of data cannot confidently answer the question of whether the microglia activation is due to a-syn aggregation directly or due to indirect effects such as neuronal dysfunction (or both), as the referee rightly asks; for this to be answered, a whole new set of work should be done (somehow blocking a-syn accumulation and seeing the effects on

microglia activation?) that goes beyond the purpose of our work. Such considerations are also discussed in the Discussion (see line 405).

As the referee requested, we also made an attempt to analyse the aggregation of a-syn in brain sections with immunofluorescence (with MJF14), to further consolidate the results obtained from the dot blot. The results obtained from the SNpc and motor cortex agree with the blot, showing accumulation of positive a-syn fibrils in the cytoplasm of some TH+ (DA) neurons and of cortical neurons from Syn rats. Additionally, the pontine nuclei showed lack of any immunofluorescence, again in line with the quantitative blot. However, immunofluorescence results in the hippocampus (str. pyramidale in CA3) and dorsal striatum were completely different from the dot blot, showing absence of a-syn accumulation in 4-month-old Syn rats, and strong background immunofluorescence also in WT animals, that make interpretations of the immunofluorescence data less reliable. Similar negative results were obtained from the striatum of 18-month-old Syn rats. Of note, the quality of the MJF14 staining for immunofluorescence was not the best, even in brain areas with positive immunofluorescence (SNpc, cortex of Syn rats). Given these, we chose to omit all the immunofluorescence data from the manuscript, leaving only the dot blot results that were much more reliable and reproducible for different animals. Below, we provide examples of some immunofluorescence images (scale bar: 20 μ m) obtained from 4-month-old rats, to make it clearer to the Reviewer.

Additional points:

1. Regarding the stereological quantification of the number of Iba1+ cells, how is possible that the total number of microglia in Figure 4B in striatum is 5000 and 10000, while in Fig 6, it is in the range of 50.000 and 100.000? (possibly a mistake in the graphs).

Reply: Indeed, this was due to a typing mistake during Figure preparation. **Figure 4B** is now corrected.

2. Interestingly the treatment did not change microglia in hippocampus. Do the authors have any hypothesis of such observation? This could be highly relevant, since the rat BAC line has been shown to have defects in hippocampal neurogenesis at this early time point, which could also be related to the microglia activation in the area. Thus, the treatment might not improve some non-motor problems in patients?

Reply: Indeed, the treatment with RvD1 was not able to reduce the numbers of microglia cells in the hippocampus of Syn rats, but it did reduce their activation level (seen as improvement, to WT levels, of morphological parameters such as number of intersections, nodes and endings or length of dendrites, see **Supplementary Fig.8**). This differential effect of RvD1 on hippocampal microglia proliferation could indeed be due to the defects in hippocampal neurogenesis but also due to many other unknown factors such as lower expression levels of hippocampal RvD1 receptors, lower affinity of RvD1 for its receptors, or receptor desensitisation due to the sub-chronic treatment, to name a few. Given that the treatment did have some effect also in the hippocampus, it is reasonable to assume that perhaps a longer treatment could be more beneficial (not excluding improvement of microglia numbers), even though in animals this wouldn't be advisable to test because the beneficial effects of the treatment could be masked by the negative effects of sub-chronic animal stress, due to the treatment protocol (repetitive i.p injections).

REVIEWERS' COMMENTS:

Reviewer #2 (Remarks to the Author):

I think the authors have address the points raised in the prior reviews. I think this manuscript will be of considerable interest to the field.

I did note that the resubmitted manuscript did not include a copy of Figure 9 (this was Figure 8 in the prior version).

Reviewer #3 (Remarks to the Author):

The authors have performed additional experiments and analysis, added the missing info and references and answered all points raised previously. The manuscript is highly improved and the data presented is significant and relevant.

We would like to thank the Reviewers for all their comments during the entire reviewing process.

Below is a reply to the last points:

Reviewer #2 (Remarks to the Author):

I think the authors have address the points raised in the prior reviews. I think this manuscript will be of considerable interest to the field.

I did note that the resubmitted manuscript did not include a copy of Figure 9 (this was Figure 8 in the prior version).

Reply: We would like to thank the Reviewer for all the comments raised during the revision of the manuscript, as the greatly improved our work. Apologies for the missing Figure.

Reviewer #3 (Remarks to the Author):

The authors have performed additional experiments and analysis, added the missing info and references and answered all points raised previously. The manuscript is highly improved and the data presented is significant and relevant.

Reply: We sincerely want to thank the Reviewer for all the comments raised during the revision of the manuscript, as we, too, believe that they helped to improve the overall quality of the work.